# RETHINKING AGAIN THE VALUE OF NETWORK PRUNING - A DYNAMICAL ISOMETRY PERSPECTIVE

## ABSTRACT

Several recent works questioned the value of inheriting weights in structured neural network pruning because they empirically found training from scratch can match or even outperform finetuning a pruned model. In this paper, we present evidence that this argument is actually *inaccurate* because of using improperly small finetuning learning rates. With larger learning rates, our results consistently suggest pruning outperforms training from scratch on multiple networks (ResNets, VGG11) and datasets (MNIST, CIFAR10, ImageNet) over most pruning ratios. To deeply understand why finetuning learning rate holds such a critical role, we examine the theoretical reason behind through the lens of *dynamical isometry*, a nice property of networks that can make the gradient signals preserve norm during propagation. Our results suggest that weight removal in pruning breaks dynamical isometry, *which fundamentally answers for the performance gap between a large finetuning LR and a small one*. Therefore, it is necessary to recover the dynamical isometry before finetuning. In this regard, we also present a regularization-based technique to do so, which is rather simple-to-implement yet effective in dynamical isometry recovery on modern residual convolutional neural networks.

## 1 INTRODUCTION

Pruning is a time-honored methodology to reduce parameters in a neural network without seriously compromising its performance (Reed, 1993; Sze et al., 2017). The prevailing pipeline of pruning comprises three steps: 1) **pretraining**: train a dense model; 2) **pruning**: prune the dense model based on certain rules; 3) **finetuning**: retrain the pruned model to regain performance. Most existing research focuses on the second step, seeking the best criterion to select unimportant weights so as to incur as less performance degradation as possible. This 3-step pipeline has been practiced for more than 30 years (Mozer & Smolensky, 1989; LeCun et al., 1990) and is still extensively adopted in today's pruning methods (Sze et al., 2017).

These said, several recent works (Crowley et al., 2018; Liu et al., 2019) questioned the necessity of inheriting weights from a pretrained model because they empirically found the small model trained from scratch can match (or sometimes outperform) the counterpart pruned from the pre-trained large model. This acutely challenges the past wisdom as well as our common belief about pruning. As far as we know, there is *no* formal response to this critical conflict. A theoretical-level understanding of this problem is even more elusive.

Meanwhile, the pruning community has been observing even more open questions. Specifically, (Renda et al., 2020; Le & Hua, 2021) found that the learning rate (LR) in finetuning holds a critical role in the final performance. A proper learning rate schedule (*e.g.*, a larger initial LR $10^{-2}$ vs. $10^{-3}$ with step-decay schedule) can improve the top-1 accuracy of a pruned ResNet-34 model (He et al., 2016) by more than $1\%$ on ImageNet (Deng et al., 2009). This discovery calls for more attention being paid to the finetuning step when comparing different pruning methods. Unfortunately, they did not present more theoretical insights to explain its occurrence. This also remains an open question in the community up to date.

In this paper, we will show these two open questions actually point to the same one. Specifically, we rerun the experiments of (Crowley et al., 2018; Liu et al., 2019) and find simply using a larger finetuning LR ($10^{-2}$ vs. $10^{-3}$ and decay it) can *significantly* improve the final performance. Compared to the improved pruning performance, training from scratch does *not* compete or surpass pruning

anymore (see Tab. 1 and Tab. 2 on ImageNet). This observation invites many questions immediately: (1) Theoretical understanding: Why does this happen? What is the theoretical reason behind it? (2) Practical solution: If this is a problem, how to fix it? Can the understanding of this problem lead us to better pruning algorithms?

This paper will present answers to all these questions. The key tool we employ to unveil the mysteries is *dynamical isometry* (Saxe et al., 2014), which describes a kind of nice property in neural networks that are easy to optimize. We carefully design an explanatory experiment using a linear MLP (multi-layer perceptron) network to demonstrate how finetuning LR affects the final performance by affecting dynamical isometry. In brief, we observe the finetuning process can recover dynamical isometry; a larger LR can help recover it faster (or better), hence the better final performance. The proposed explanation is validated by our empirical results and resonates with many empirical observations. Furthermore, by the explanation, we learn dynamical isometry recovery is rather imperative. To achieve so, we present a very simple regularization-based method for pruning and show its effectiveness in recovering dynamical isometry on modern residual convolutional neural networks (CNNs).

**Contributions**. (1) We empirically demonstrate the questioning about the value of inheriting weights in structured pruning in previous works is inaccurate and point out that the direct cause is improperly using a small finetuning LR. Our finding justifies the value of inheriting weights in structured pruning. (2) On top of the empirical finding, more importantly, we present a theoretical explanation through examining the dynamical isometry of networks in pruning. This explanation is empirically validated by our carefully designed control experiments. (3) In addition to the theoretical understanding, we also propose a regularization-based method for dynamical isometry recovery. Despite its brutal simplicity, it is shown effective to recover the broken dynamical isometry on modern residual convolutional neural networks.

## 2 RELATED WORK

**Conventional pruning**. Pruning aims to remove as many parameters as possible in a neural network meanwhile maintaining its performance. There are many ways to categorize pruning methods. The most popular two are grouping by pruning structure and methodology.

(1) In terms of pruning structure, pruning can be specified into unstructured pruning (Han et al., 2015; 2016) and structured pruning (Wen et al., 2016; Li et al., 2017; He et al., 2017). For the former, a single weight is the basic pruning element. Unstructured pruning can deliver a high compression ratio; whereas, without regularization, the pruned locations usually spread *randomly* in the network, which is hard to exploit for acceleration. On the opposite, structured pruning introduces certain patterns in the pruned locations, which benefit subsequent acceleration while cannot achieve as much compression. Choices between unstructured and structured pruning depend on specific application needs. For structured pruning, there are still many sub-groups (Mao et al., 2017). In the literature, without specific mention, structured pruning means filter pruning or channel pruning. This paper focuses on **structured (filter) pruning** because the "no value of inheriting weights" argument is mainly discussed in this context (Liu et al., 2019).

(2) In terms of pruning methodology (*i.e.*, how to select unimportant weights to prune), pruning falls into two paradigms in general: importance-based and penalty-based. The former prunes weights based on some established importance criteria, such as magnitude (for unstructured pruning) (Han et al., 2015; 2016) or $L_1$-norm (for filter pruning) (Li et al., 2017), saliency based on 2nd-order gradients (*e.g.*, Hessian or Fisher) (LeCun et al., 1990; Hassibi & Stork, 1993; Theis et al., 2018; Wang et al., 2019a; Singh & Alistarh, 2020). The latter adds a penalty term to the objective function, drives unimportant weights towards zero, then removes those with the smallest magnitude. Note, the two groups are *not* starkly separated. Many methods take wisdom from both sides. For example, (Ding et al., 2018; Wang et al., 2019b; 2021b) select unimportant weights by magnitude (akin to the first group) while also employing the regularization to penalize weights (akin to the second group). There is no conclusion about which paradigm is better, yet empirically, the state-of-the-art pruning methods are closer to the second paradigm, *i.e.*, deciding weights via training instead of some derived formulas. Although no theories have formally discussed the reason, we can take a rough guess with the knowledge from this paper: Training can recover dynamical isometry, which is beneficial to subsequent finetuning.

For more comprehensive literature, we refer interested readers to several surveys: an outdated one (Reed, 1993), some recent surveys of pruning alone (Gale et al., 2019; Blalock et al., 2020) or pruning as a sub-topic under the general umbrella of model compression and acceleration (Sze et al., 2017; Cheng et al., 2018a;b; Deng et al., 2020).

**Pruning at initialization (PaI)**. Recent years have seen several new pruning paradigms. The most prominent one is pruning at initialization. Different from the conventional pruning, which prunes a *pretrained* model, PaI methods prune a *randomly initialized* model. Existing PaI approaches mainly include (Lee et al., 2019; 2020; Wang et al., 2020; Frankle et al., 2021; Ramanujan et al., 2020) and the series of lottery ticket hypothesis (Frankle & Carbin, 2019; Frankle et al., 2020). Interested readers may refer to (Wang et al., 2021a) for a comprehensive summary about PaI.

This topic is relevant to this work mainly because one PaI paper (Lee et al., 2020) also examines pruning using the tool of dynamical isometry. The similarity between our paper and theirs is that we both employ dynamical isometry as a tool to examine the property of network pruning. However, our paper is *significantly different* from theirs in many axes: (1) Basic setting. The most obvious difference is that we focus on pruning a *pretrained* model while (Lee et al., 2020) focuses on pruning at initialization (PaI). They are two different tracks in pruning (as such, PaI methods typically do *not* compare with the methods of pruning pretrained models) and the latter was shown to consistently underperform the former (Frankle et al., 2021; Wang et al., 2021a). (2) Motivation. Despite the same tool (mean JSV), (Lee et al., 2020) uses it to select unimportant weights to prune (*i.e.*, for a new pruning criterion), while we use it to analyze why finetuning LR has a significant impact on final performance. The role of finetuning LR in pruning is *not* mentioned at all in their paper. (3) Proposed technical method. (Lee et al., 2020) focuses on *unstructured pruning*, while we focuses on *structured pruning*. This further leads to fundamental difference when designing the dynamical isometry recovery (DIR) methods – In (Lee et al., 2020), their proposed method is to use iterative optimization for *approximated* isometry (due to the irregular sparsity); while in our case, since the pruned filers can be completely removed from the network, one of our DIR method (OrthP) has *closed-form* solution and can achieve *exact* isometry. (4) Finally, in terms of empirical results, (Lee et al., 2020) only conducts experiments on MNIST (LeCun et al., 1998) and CIFAR (Krizhevsky, 2009), while we have extensive results on the large-scale ImageNet dataset (Deng et al., 2009).

## 2.1 EMPIRICAL STUDY: LARGER FINETUNING LR IS CRITICAL

As far as we know, mainly *two* papers question the value of inheriting weights from a pretrained model: (Crowley et al., 2018; Liu et al., 2019). Both papers draw two similar conclusions. (1) Inheriting weights from a pretrained model in pruning has *no* value, *i.e.*, training from scratch the small model can match (or outperform sometimes) the counterpart pruned from a big pretrained model. (2) Given the fact of (1), what really matters in pruning may lie in the pruned *architecture* instead of the inherited weight values. As such, both papers propose to view pruning as a form of neural architecture search (Zoph & Le, 2017; Elsken et al., 2019). In this section, we first reexamine the empirical studies in (Crowley et al., 2018; Liu et al., 2019) to show that the "no value of inheriting weights" argument is actually inaccurate owing to improper finetuning LR schedules.

**Reexamination of (Liu et al., 2019)**. Before presenting results, here are some important comparison setting changes worth particular attention: (1) In (Liu et al., 2019), they compare training from scratch with *six* pruning methods (five structured pruning methods (Li et al., 2017; Luo et al., 2017; Liu et al., 2017; He et al., 2017; Huang & Wang, 2018) and one unstructured pruning method (Han et al., 2015)). Here, we only focus on the $L_1$-*norm pruning* (Li et al., 2017) on ImageNet. The main reason is that, $L_1$-norm pruning is well-known a very *basic* filter pruning method. If we can show it outperforms training from scratch already, it will be no surprise to see other more advanced pruning methods also outperform training from scratch. In this sense, $L_1$-norm pruning is the most representative method here for our investigation. (2) In (Liu et al., 2019), they have two variants for the number of epochs in scratch training, "Scratch-E" and "Scratch-B". For the former, different small models are trained for a fixed number of epochs; for the latter, *smaller* models are trained for *more* epochs to maintain the same computation budget (Scratch-B was shown to be better than Scratch-E in (Liu et al., 2019)). Also, they decay LR only to $10^{-3}$ following the official PyTorch ImageNet example[1]. Here, we simply train all the networks for the same number of epochs but

---

[1] https://github.com/pytorch/examples/tree/master/imagenet

Table 1: Top-1 accuracy comparison of different implementations of the $L_1$-norm pruning (Li et al., 2017) on ImageNet. We adopt the torchvision models as unpruned models for fair comparison. ResNet-34-A speedup: $1.18\times$. ResNet-34-B speedup: $1.32\times$. The results of (Li et al., 2017) and (Liu et al., 2019) are directly cited from their papers. The best cases in our training from scratch and pruning are randomly repeated for 3 times ($\pm$ indicates stddev) to prevent random variation.

| Implementation | Unpruned (%) | Pruned model | Scratch (%) | Pruned-Finetuned (%) | Finetuning LR schedule |
|---|---|---|---|---|---|
| (Li et al., 2017) | 73.23 | ResNet-34-A | (Not reported) | 72.56 | 20 epochs, initial $10^{-3}$, fixed |
| | | ResNet-34-B | (Not reported) | 72.17 | 20 epochs, initial $10^{-3}$, fixed |
| (Liu et al., 2019) | 73.31 | ResNet-34-A | **73.03** | 72.56 | 20 epochs, initial $10^{-3}$, fixed |
| | | ResNet-34-B | **72.91** | 72.29 | 20 epochs, initial $10^{-3}$, fixed |
| Our rerun | 73.31 | ResNet-34-A | $73.51_{\pm 0.12}$ | 72.91 | 20 epochs, initial $10^{-3}$, fixed |
| | | | | 72.94 | 90 epochs, initial $10^{-3}$, fixed |
| | | | | 73.88 | 90 epochs, initial $10^{-3}$, decay |
| | | | | $\mathbf{73.92_{\pm 0.03}}$ | 90 epochs, initial $10^{-2}$, decay |
| Our rerun | 73.31 | ResNet-34-B | $73.16_{\pm 0.12}$ | 72.50 | 20 epochs, initial $10^{-3}$, fixed |
| | | | | 72.58 | 90 epochs, initial $10^{-3}$, fixed |
| | | | | 73.61 | 90 epochs, initial $10^{-3}$, decay |
| | | | | $\mathbf{73.62_{\pm 0.04}}$ | 90 epochs, initial $10^{-2}$, decay |

Table 2: Top-1 accuracy comparison between scratch training ("Scratch") and $L_1$-norm pruning (Li et al., 2017) on ImageNet. "PR" means pruning ratio. $^\dagger$We adopt the official torchvision models as unpruned models. "Finetuned-1" and "Finetuned-2" refers to two finetuning LR schedules ("Finetuned-1": 90 epochs, initial $10^{-3}$, decay 30/60/75; "Finetuned-2": 90 epochs, initial $10^{-2}$, decay 45/68). Best results are in **bold**, second best underlined, each averaged by 3 random runs.

| Network | PR | Params reduc. (%) | FLOPs reduc. (%) | Scratch (%) | Pruned-Finetuned-1 (%) | Pruned-Fintuned-2 (%) |
|---|---|---|---|---|---|---|
| ResNet-18 | 0 | 0 | 0 | $69.76^\dagger$ | / | / |
| | 0.1 | 9.56 | 9.58 | $70.15_{\pm 0.02}$ | $\underline{70.43_{\pm 0.02}}$ | $\mathbf{70.48_{\pm 0.10}}$ |
| | 0.3 | 28.32 | 28.18 | $68.90_{\pm 0.10}$ | $\underline{69.29_{\pm 0.07}}$ | $\mathbf{69.54_{\pm 0.09}}$ |
| | 0.5 | 47.03 | 46.20 | $67.03_{\pm 0.01}$ | $\underline{67.36_{\pm 0.03}}$ | $\mathbf{67.71_{\pm 0.05}}$ |
| | 0.7 | 65.99 | 64.93 | $64.21_{\pm 0.10}$ | $\underline{63.72_{\pm 0.03}}$ | $\mathbf{64.45_{\pm 0.04}}$ |
| | 0.9 | 84.75 | 83.52 | $\mathbf{56.70_{\pm 0.17}}$ | $53.49_{\pm 0.10}$ | $\underline{55.89_{\pm 0.11}}$ |
| | 0.95 | 89.51 | 88.03 | $\mathbf{51.83_{\pm 0.14}}$ | $44.46_{\pm 0.15}$ | $\underline{49.99_{\pm 0.10}}$ |
| ResNet-34 | 0 | 0 | 0 | $73.31^\dagger$ | / | / |
| | 0.1 | 9.84 | 9.92 | $73.53_{\pm 0.10}$ | $\underline{73.86_{\pm 0.05}}$ | $\mathbf{74.04_{\pm 0.05}}$ |
| | 0.3 | 29.15 | 29.26 | $72.50_{\pm 0.17}$ | $\underline{73.11_{\pm 0.06}}$ | $\mathbf{73.31_{\pm 0.08}}$ |
| | 0.5 | 48.41 | 48.12 | $71.27_{\pm 0.03}$ | $\underline{71.71_{\pm 0.06}}$ | $\mathbf{71.85_{\pm 0.07}}$ |
| | 0.7 | 67.95 | 67.63 | $68.69_{\pm 0.10}$ | $\underline{68.90_{\pm 0.08}}$ | $\mathbf{69.33_{\pm 0.04}}$ |
| | 0.9 | 87.26 | 86.97 | $62.08_{\pm 0.12}$ | $\underline{60.34_{\pm 0.03}}$ | $\mathbf{62.26_{\pm 0.06}}$ |
| | 0.95 | 92.16 | 91.69 | $\mathbf{57.21_{\pm 0.15}}$ | $52.83_{\pm 0.11}$ | $\underline{56.69_{\pm 0.23}}$ |
| VGG11_BN | 0 | 0 | 0 | $70.37^\dagger$ | / | / |
| | 0.1 | 9.19 | 17.78 | $68.51_{\pm 0.04}$ | $\underline{71.45_{\pm 0.07}}$ | $\mathbf{71.74_{\pm 0.04}}$ |
| | 0.3 | 26.80 | 47.63 | $66.60_{\pm 0.11}$ | $\underline{70.00_{\pm 0.03}}$ | $\mathbf{70.53_{\pm 0.05}}$ |
| | 0.5 | 43.86 | 70.56 | $65.85_{\pm 0.13}$ | $\underline{67.34_{\pm 0.05}}$ | $\mathbf{68.01_{\pm 0.10}}$ |
| | 0.7 | 60.54 | 87.03 | $61.56_{\pm 0.06}$ | $\underline{62.14_{\pm 0.09}}$ | $\mathbf{63.14_{\pm 0.08}}$ |
| | 0.9 | 76.50 | 96.49 | $\underline{48.34_{\pm 0.12}}$ | $45.11_{\pm 0.07}$ | $\mathbf{48.52_{\pm 0.06}}$ |
| | 0.95 | 80.49 | 97.76 | $\underline{35.47_{\pm 0.09}}$ | $33.50_{\pm 0.10}$ | $\mathbf{38.47_{\pm 0.13}}$ |

ensure the epochs are abundant (120 epochs) and decay LR to a very small amount ($10^{-5}$). These two changes are to make sure the networks are trained to *full convergence*. As we will show, one primary cause possibly leading (Liu et al., 2019) to an inaccurate conclusion is exactly that the pruned networks are *not* fully converged (see Tab. 1).

With the LR schedule changes, we rerun the experiments *using the released code of (Liu et al., 2019)*. Results are presented in Tab. 1. In the implementations of (Liu et al., 2019), the finetuned model is outperformed by the scratch training one, hence their "no value of inheriting weights" argument. We also reproduce their settings (the two rows of "20 epochs, initial $10^{-3}$, fixed" in "Our rerun") for confirming their argument. However, the finetuning LR schedule "20 epochs, initial $10^{-3}$, fixed" is actually sub-optimal; the network is *not* fully converged. Using the proper ones ("90 epochs, initial $10^{-3}$, decay" or "90 epochs, initial $10^{-2}$, decay"), pruning *outperforms* training from

scratch for both ResNet-34-A and ResNet-34-B. (We note the pruned models even outperform the original models. This is probably because pruning reduces the network redundancy, thus curbing overfitting. This phenomenon is also widely observed in past pruning works (Han et al., 2016; Wen et al., 2016; He et al., 2017) especially under small pruning ratios as in Tab. 1.)

Tab. 1 only presents two ResNet models and their speedups are actually quite small. To see if the finetuning LR effect still holds across the full spectrum of pruning ratios and on other types of networks, we vary the pruning ratios from 0.1 to 0.95 and include experiments on VGG11_BN (Simonyan & Zisserman, 2015).

Results are presented in Tab. 2. With a more proper finetune LR scheme (column "Pruned-Fintuned-2" vs. "Pruned-Fintuned-1"), the performance can be improved *significantly*. A clear pattern is, the larger the pruning ratio, the more of the improvement. Now, comparing the results of "Pruned-Fintuned-2" to those of "Scratch", we can see pruning *outperforms* scratch-training in most cases. Exceptions appear on ResNet-34/18 under extreme pruning ratios (90% and 95%). Despite them, we believe it is fair to say **inheriting weights *has* value given the fact that 17/20 experiments in Tabs. 1 and 2 show pruning is better than training from scratch**, especially under the pruning ratios of practical interest (*i.e.*, non-extreme pruning ratios). Retrospectively, (Liu et al., 2019) concluded oppositely because they faithfully re-implemented the $L_1$-norm pruning method *just according to the description in the original paper* (Li et al., 2017): fixed LR $10^{-3}$, 20 epochs, which turns out *far from optimal* as we know now.

**Reexamination of (Crowley et al., 2018)**. Coincidentally, (Crowley et al., 2018) adopted a *very similar* finetuning LR scheme to (Liu et al., 2019): They finetuned the pruned network with the lowest LR ($8 * 10^{-4}$, close to $10^{-3}$ in (Liu et al., 2019)) during scratch training and also *fixed*. Like the empirical study above, we reproduce the experiments of (Crowley et al., 2018) and rerun them with a larger initial LR ($10^{-2}$) and decay it during finetuning.

Detailed results are deferred to the Appendix (Tab. 10) due to the limited length here. We summarize the observation here – Exactly the same as the case in (Liu et al., 2019), when the proper finetuning LR is used, pruning actually *outperforms* the best scratch training scheme consistently.

Up to now, the results above have shown that the "no value of inheriting weights" argument in previous works is largely attributed to sup-optimal finetuning settings. A larger LR (*e.g.*, $10^{-2}$) can significantly improve the finetuning performance than a small one (*e.g.*, $10^{-3}$). In fact, we are not the only one to discover this. Previous works (Renda et al., 2020; Le & Hua, 2021) also reported similar observation. Nevertheless, they do not link the phenomenon with the "value of inheriting weights" argument and do not conduct systematical empirical studies as we do here. *More importantly, neither of them presented theoretical explanations about its occurrence* – next, we are about to bridge this gap. We present a faithful theoretical explanation through the lens of dynamical isometry.

## 3 DYNAMICAL ISOMETRY IS THE KEY

### 3.1 PREREQUISITE: DYNAMICAL ISOMETRY

Dynamical isometry (DI) is studied under the topic of trainability of deep neural networks. It was first brought up in (Saxe et al., 2014). Specifically, the dynamical isometry is defined as *the singular values of the Jacobian matrix being around 1* (Saxe et al., 2014). It is easy to see that the networks with dynamical isometry is easy to train, because JSVs around 1 imply the gradient signals will not be amplified or attenuated seriously during propagation, preventing the network from gradient exploding or vanishing, which are well-known the main difficulties in deep network training (Glorot & Bengio, 2010; Sutskever et al., 2013). For linear networks, dynamical isometry can be achieved *exactly* by the orthogonal initialization proposed in (Saxe et al., 2014); while for neural networks with non-linearity (like ReLU (Nair & Hinton, 2010)) and convolution, it can only be approximated up to date (see Tab. 3).

**Mean Jacobian singular values**. Since dynamical isometry is measured by the Jacobian singular values (JSV's), we adopt *the mean of Jacobian singular values* (denoted by $\bar{S}$) as a scalar metric for analysis. Specifically, for a Jacobian $\mathbf{J} \in \mathbb{R}^{C \times D_{\text{in}}}$ ($C$ stands for the output dimension, *i.e.*, the number of classes, $D_{\text{in}}$ for the input dimension), apply singular value decomposition (Trefethen &

Table 3: JSVs (Jacobian singular values) of orthogonal initialization (Saxe et al., 2014) on different types of neural networks on MNIST dataset. Note, only the linear MLP network can achieve dynamical isometry *exactly* (*i.e.*, all the JSVs equal to 1).

| Network | Mean JSV | Max JSV | Min JSV |
|---|---|---|---|
| MLP-7-Linear | $1.0000_{\pm 0.0000}$ | 1.0000 | 1.0000 |
| MLP-7-ReLU | $1.2268_{\pm 0.5519}$ | 3.2772 | 0.2282 |
| LeNet-5-Linear | $0.9983_{\pm 0.0842}$ | 1.2330 | 0.7896 |
| LeNet-5-ReLU | $1.8331_{\pm 0.5731}$ | 3.6007 | 0.6151 |

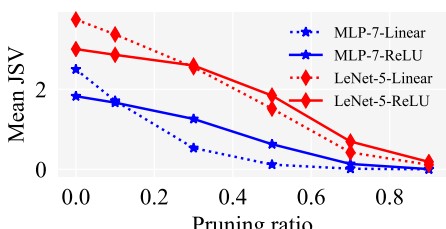

Figure 1: Mean JSV (Eq. (1)) of pruned networks w.r.t. different pruning ratios on the MNIST dataset. Note, with a larger pruning ratio, mean JSV is hurt more.

Bau III, 1997) to it,

$$U, \Sigma, V = \text{svd}(\mathbf{J}), \bar{S} = \frac{1}{K} \sum_{i=1}^{K} \Sigma_{ii}, \tag{1}$$

where $\Sigma$ is the singular value matrix and $K = \min(C, D_{\text{in}})$.

**Pruning as *poor* initialization**. We investigate how the pruning affects $\bar{S}$. The results are shown in Fig. 1. As seen, the mean JSV is consistently *damaged* by pruning; and a larger pruning ratio, more decrease of the mean JSV. This means, pruning actually servers as a very *poor* initialization scheme for the subsequent finetuning.

In stark contrast to the broad awareness that initialization is rather critical to neural network training (Glorot & Bengio, 2010; Sutskever et al., 2013; Mishkin & Matas, 2016; Krähenbühl et al., 2015; He et al., 2015), the initialization role of pruning has received negligible research attention, however. As far as we know, no prior works have noted this issue when pruning a pretrained network or tried to recover the broken dynamical isometry before finetuning.

Simply put, our goal next is to show it is this broken dynamical isometry that answers for the performance gap between LR $10^{-2}$ and $10^{-3}$. If dynamical isometry is fully recovered before finetuning, LR $10^{-2}$ vs. $10^{-3}$ should not cause significant performance difference anymore. To validate this explanation, we need a method to *fully recover* dynamical isometry in filter pruning, as is introduced next.

### 3.2 DYNAMICAL ISOMETRY RECOVERY IN FILTER PRUNING

In (Saxe et al., 2014), they propose a weight orthogonalization scheme to achieve dynamical isometry for neural network *initialization*, namely, the initial weights are *randomly sampled*. Different from their case, here the initial weights are inherited from a pretrained model by pruning. Therefore, we need to adapt it to our application.

For a fully-connected layer parameterized by a matrix $W_0 \in \mathbb{R}^{J \times K}$ (for a convolutional layer parameterized by a 4-d tensor of shape $\mathbb{R}^{N \times C \times H \times W}$, it can be reshaped to a matrix of shape $\mathbb{R}^{N \times CHW}$), it reduces to matrix $W$ of size $\mathbb{R}^{J_1 \times M_1}$ ($J_1 \leq J, K_1 \leq K$) after structured pruning. Then, we apply the weight orthogonalization technique (Mezzadri, 2006) based on QR decomposition (Trefethen & Bau III, 1997) to $W$,

$$\begin{aligned} Q, R &= \text{qrd}(W), \\ W^* &= Q \odot \text{sign}(\text{diag}(R)), \end{aligned} \tag{2}$$

where qrd($\cdot$) means QR decomposition; $Q$ is an orthogonal matrix of the same size as $W$ ($\mathbb{R}^{J_1 \times K_1}$); $R$ is an an upper triangular matrix of size $\mathbb{R}^{K_1 \times K_1}$; sign($\cdot$) is the sign function which returns the positive or negative sign of its argument; $\odot$ represents the Hadamard (element-wise) product aligned to the last axis (since $Q$ and sign(diag(R)) share the same dimension at the last axis).

As an orthogonalized version of $W$, $W^*$ recovers the dynamical isometry damaged by pruning. Therefore, we propose to employ $W^*$ instead of the original $W$ as the initialization weights for later finetuning. We dub this weight orthogonalization method for *pruned* models as **OrthP**. With this method, we can fully recover broken dynamical isometry and continue our analysis as follows.

Table 4: Mean JSV of the first 10 epochs under different finetuning settings. Epoch 0 refers to the model just pruned, before any finetuning. Pruning ratio is 0.9. Note, with OrthP, the mean JSV is 1 because OrthP can achieve exact isometry.

| Epoch | 0 | 1 | 2 | 3 | 4 | 5 | 6 | 7 | 8 | 9 | 10 |
|---|---|---|---|---|---|---|---|---|---|---|---|
| LR=$10^{-2}$, w/o OrthP | 0.0004 | 0.6557 | 0.8946 | 1.0191 | 0.9826 | 1.0965 | 1.1253 | 1.2595 | 1.3298 | 1.2940 | 1.4238 |
| LR=$10^{-3}$, w/o OrthP | 0.0004 | 0.0004 | 0.0006 | 0.0014 | 0.1103 | 0.2765 | 0.3501 | 0.4320 | 0.5167 | 0.7478 | 0.8501 |
| LR=$10^{-2}$, w/ OrthP | 1.0000 | 1.2318 | 1.4144 | 1.4277 | 1.4017 | 1.4709 | 1.5171 | 1.5551 | 1.6082 | 1.6538 | 1.6648 |
| LR=$10^{-3}$, w/ OrthP | 1.0000 | 1.5135 | 1.6630 | 1.7449 | 1.8250 | 1.8720 | 1.9193 | 1.9556 | 1.9943 | 2.0084 | 2.0409 |

Table 5: Test accuracy (%) of the first 10 epochs *corresponding to Tab. 4* under different finetuning settings. Epoch 0 refers to the model just pruned, before any finetuning. Pruning ratio is 0.9.

| Epoch | 0 | 1 | 2 | 3 | 4 | 5 | 6 | 7 | 8 | 9 | 10 |
|---|---|---|---|---|---|---|---|---|---|---|---|
| LR=$10^{-2}$, w/o OrthP | 9.74 | 63.86 | 79.96 | 79.74 | 80.06 | 85.79 | 85.82 | 86.11 | 86.45 | 86.53 | 85.95 |
| LR=$10^{-3}$, w/o OrthP | 9.74 | 9.74 | 9.74 | 12.09 | 21.74 | 27.95 | 33.55 | 35.92 | 49.19 | 65.50 | 69.90 |
| LR=$10^{-2}$, w/ OrthP | 9.74 | 91.05 | 91.39 | 91.33 | 91.37 | 91.74 | 91.69 | 90.74 | 91.39 | 91.58 | 91.44 |
| LR=$10^{-3}$, w/ OrthP | 9.74 | 90.81 | 91.59 | 91.77 | 91.85 | 92.04 | 92.12 | 92.22 | 92.12 | 92.33 | 92.25 |

### 3.3 ANALYSIS WITH MLP-7-LINEAR ON MNIST

**Evaluated network**. The network for analysis is a 7-layer linear MLP. We are aware that this toy network has little practical meaning, but it is very appropriate here for two reasons. First, as mentioned above, in our analysis we need a method to recover DI *exactly*. Up to date, this can *only* be achieved on linear networks (see Tab. 3). Second, the linear MLP network is free from the intervention of modern CNN features (*e.g.*, BN (Ioffe & Szegedy, 2015), residual (He et al., 2016)). By our observation, these features will make the problem complex and prevent us from seeing consistent results at the early analysis stage.

**LR schedule setup**. When we set different LR schedules with different initial LRs, we will (1) keep the total number of epochs the same, (2) keep the last LR the same, (3) intentionally use *prolonged* training epochs (like 90 or even 900 epochs on MNIST dataset. Typically, it only needs 30 epochs to reach convergence on MNIST). All of these are to ensure the networks are fully converged, helping us render a faithful conclusion. Step decay LR schedule is employed given its broad use.

**Proposed explanation and its deducted hypotheses**. We list the first 10-epoch mean JSV and test accuracy of pruning MLP-7-Linear (at pruning ratio 0.9) under different finetuning setups in Tab. 4 and Tab. 5, respectively. The following observations are worth our attention.

- In Tab. 4, one important fact is that, the mean JSV can recover itself without any extra help during finetuning, regardless of different setups.

- In Tab. 4, without OrthP, the mean JSV of LR $10^{-2}$ arises *much faster* than that of LR $10^{-3}$. By our analysis in the Appendix (Sec. D), a larger mean JSV implies better dynamical isometry, which further implies easier optimization. Easier optimization finally leads to the better test accuracy of LR $10^{-2}$ against $10^{-3}$ when the training epoch is insufficient, as shown in Tab. 5.

- In Tab. 4, with OrthP, the broken dynamical isometry is exactly recovered (note at epoch 0, the mean JSV is 1). Then the accuracy advantage of LR $10^{-2}$ over $10^{-3}$ is not significant anymore, *e.g.*, in Tab. 5, at epoch 10, LR $10^{-2}$ is better than LR $10^{-3}$ by 16.05% *without* OrthP; while it is not better (actually worse) than LR $10^{-3}$ *with* OrthP.

- Particularly note how the mean JSV trend in Tab. 4 correlates well with the test accuracy trend in Tab. 5. This promotes the plausible idea that it is dynamical isometry that answers for the accuracy gap fundamentally.

These observations inspire us to the following plausible explanation about how a larger finetuning LR can improve performance significantly:

> *A larger finetuning LR helps the network update faster, thus the dynamical isometry (measured by mean JSV) recovers faster (and possibly better), which further leads to faster (and possibly better) optimization. For deep networks nowadays, better optimization typically implies better generalization, thus the larger finetuning LR eventually leads to the better test accuracy as we see.*

Table 6: Summary of finetuning LR setups corresponding to the 4 proposed hypotheses in Sec. 3.3.

|  | Initial LR $10^{-2}$ | Initial LR $10^{-3}$ |
|---|---|---|
| For Hypothesis 1 | 90 epochs, Initial $10^{-2}$, decay 30/60 | 90 epochs, Initial $10^{-3}$, decay 45 |
| For Hypothesis 2 | 900 epochs, Initial $10^{-2}$, decay 300/600 | 900 epochs, Initial $10^{-3}$, decay 450 |
| For Hypothesis 3 | OrthP, 90 epochs, Initial $10^{-2}$, decay 30/60 | OrthP, 90 epochs, Initial $10^{-3}$, decay 45 |
| For Hypothesis 4 | OrthP, 900 epochs, Initial $10^{-2}$, decay 300/600 | OrthP, 900 epochs, Initial $10^{-3}$, decay 450 |

That is, a larger finetuning LR shows performance advantage *only if dynamical isometry is broken by pruning first*. If dynamical isometry is fully recovered before finetuning, a larger LR should *not* pose performance advantage anymore. The validation of this explanation can be specified into the following 4 deducted hypotheses:

- **Hypothesis 1**: Given a small number of epochs, mean JSV *cannot* be fully recovered by training, then the larger LR should show a significant advantage over the smaller LR.

- **Hypothesis 2**: With sufficient training epochs, mean JSV can be fully recovered by training. Then the larger LR should have less advantage over the smaller LR.

- **Hypothesis 3**: If we employ OrthP to exactly recover the mean JSV, given the small number of epochs again, the larger LR should have much less advantage now.

- **Hypothesis 4**: If we combine abundant epochs with OrthP, mean JSV will be recovered even completely, then the performance advantage of the larger LR over the smaller one should be even weaker.

Corresponding to these four hypotheses, the eight finetuning LR settings are summarized in Tab. 15. The unpruned MLP model is trained with LR schedule "90 epochs, initial $10^{-2}$, decay 30/60". For filter pruning, we employ $L_1$-norm pruning (Li et al., 2017) throughout this paper. Specifically, it sorts the neurons (or filters) by their $L_1$ norms in ascending order and prunes those with the least norms by a predefined pruning ratio $r$.

The final accuracy results are shown in Tab. 7. We first analyze the results of pruning ratio 0.8 in Tab. 7. As seen, when finetuned for 90 epochs, LR $10^{-2}$ shows an advantage over LR $10^{-3}$ by 0.82% accuracy. It is tempting to draw a conclusion based on this comparison that LR $10^{-2}$ is much better than LR $10^{-3}$. However, this is not the whole story:

- With 900 epochs, LR $10^{-2}$ is greatly surpassed by LR $10^{-3}$ (91.64 vs. 92.54). The reason by our proposed explanation is that, with abundant epochs, the dynamical isometry can be recovered more completely, hence LR $10^{-2}$ does not show advantages anymore over $10^{-3}$.

- When OrthP applied, LR $10^{-2}$ does not show significant advantages either, similar to the effect of increasing the number of training epochs. This is because that finetuning shares the same role of recovering dynamical isometry with OrthP. Just OrthP is more effective since it is analytically targeting exact dynamical isometry.

- When the best setting used (OrthP + 900 epochs), LR $10^{-3}$ is slightly better than $10^{-2}$. Comparing "OrthP, 900 epochs" with "OrthP, 90 epochs", the gains are only marginal. This is because the dynamical isometry has already been fully recovered by OrthP, thus more training epochs do not show much value anymore.

A different pruning ratio 0.9 is also explored. Its results are in line with those of pruning ratio 0.8 as shown in Tab. 7. In short, these empirical observations are *fully in line with* our expectations, justifying the validity of the proposed explanation.

**Explaining LR effect on ImageNet**. In Tab. 2, there is an apparent trend that the *larger* pruning ratio, the *more* performance advantage of LR $10^{-2}$ over $10^{-3}$ (this also appears on CIFAR10 dataset, see Tab. 8). Using our explanation, this phenomenon can also be explained now – When the pruning ratio is greater, more dynamical isometry is damaged. LR $10^{-2}$ can find more use in these cases since it is faster/better to recover dynamical isometry, hence the more pronounced advantage.

**A closer look and more lessons**. In the ResNet34 results above (Tab. 1) reported by (Li et al., 2017) and (Liu et al., 2019)), it is still easy to notice their finetuning LR schedule may be sub-optimal given

Table 7: Test accuracies (%) of the 4 hypotheses in Tab. 15 with **MLP-7-Linear** network on MNIST. Accuracy of unpruned model: 92.77%. Each setting is randomly run 5 times, mean accuracy and stddev reported. "Acc. gain" refers to the mean accuracy improvement of LR $10^{-2}$ over $10^{-3}$. Hyper-parameters: batch size 100, SGD optimizer, weight decay 0.0001, LR schedule: initial LR 0.01, decayed at epoch 30 and 60 by factor 0.1, total epochs: 90. The LR schedule is used for both scratch training and finetuning.

| Finetuning setting | Pruning ratio 80% | | | Pruning ratio 90% | | |
|---|---|---|---|---|---|---|
| | LR $10^{-3}$ | LR $10^{-2}$ | Acc. gain | LR $10^{-3}$ | LR $10^{-2}$ | Acc. gain |
| 90 epochs | $90.54_{\pm 0.02}$ | $\mathbf{91.36_{\pm 0.02}}$ | 0.82 | $87.59_{\pm 0.01}$ | $\mathbf{87.81_{\pm 0.03}}$ | 0.22 |
| 900 epochs | $\mathbf{92.54_{\pm 0.03}}$ | $91.64_{\pm 0.41}$ | $-0.90$ | $\mathbf{90.44_{\pm 0.01}}$ | $87.83_{\pm 0.04}$ | $-2.61$ |
| OrthP, 90 epochs | $92.77_{\pm 0.03}$ | $\mathbf{92.79_{\pm 0.03}}$ | 0.02 | $92.72_{\pm 0.03}$ | $\mathbf{92.77_{\pm 0.04}}$ | 0.05 |
| OrthP, 900 epochs | $\mathbf{92.84_{\pm 0.03}}$ | $92.81_{\pm 0.04}$ | $-0.03$ | $\mathbf{92.86_{\pm 0.03}}$ | $92.79_{\pm 0.03}$ | $-0.07$ |
| Scratch (Kaiming uniform) | | $92.60_{\pm 0.14}$ | | | $91.48_{\pm 0.23}$ | |
| Scratch (Orthogonal init) | | $92.76_{\pm 0.03}$ | | | $92.76_{\pm 0.04}$ | |

their finetuning process (20 epochs) is obviously shorter than training from scratch (90 epochs). For Tab. 7, quite differently, the "90 epochs" finetuning setting appears *nothing wrong* (considering the unpruned network is trained for 90 epochs, finetuning for 90 epochs is not short; besides, the LR is also decayed to a very small amount). However, the results of this setting actually lead us to a partial conclusion that the larger LR is better than the smaller LR. This particular example shows *how misleading the comparison results can be, even though they look perfectly fair, if we are not aware of the effect of dynamical isometry in structured pruning.*

## 4    Dynamical Isometry Recovery (DIR)

By our analysis above, dynamical isometry *recovery* before finetuning is rather important. An effective dynamical isometry recovery method is supposed to (1) close up the performance gap between finetuning LR $10^{-2}$ and $10^{-3}$, (2) improve the final performance (otherwise, there is no point using it at all). As shown in Tab. 7, the proposed OrthP method for linear MLP networks can meet these two requirements; and since it recovers DI exactly, it actually completely closes the performance gap between finetuning LR $10^{-2}$ and $10^{-3}$. Unfortunately, this method does not generalize to more practical non-linear convolutional neural networks, as shown in Tab. 8, where using OrthP *degrades* the final performance. As a remedy, in this section we introduce a very simple regularization-based method that can work on practical CNNs.

**Strong $L_2$ regularization as a drop-in remedy**. In a typical structured pruning algorithm, unimportant filters are selected by some criterion. Then they are removed (zeroed out), followed by a finetuning process. We propose to apply a *super strong $L_2$* regularization (*e.g.*, regularization factor equals to 1) to push the unimportant filters to rather close to zero first, before permanently removing them. This simple technique can readily work as a drop-in step into *any* structured pruning algorithm and it is very easy to implement on any deep learning framework.

In Appendix (Sec. E), we provide more explanation regarding how StrongReg is related to dynamical isometry.

Results in Tab. 8 demonstrate the effectiveness of this simple technique on ResNet56. It improves the final pruning performance and also shrinks the performance gap between finetuning LR $10^{-2}$ and $10^{-3}$ (especially when the pruning ratio is large), meeting the two requirements above. We are aware that this technique is preceded by several variants in the past literature (*e.g.*, GReg-1 in (Wang et al., 2021b)), but they do not link their method with dynamical isometry recovery as we do here. Also, our technique is even simpler than theirs – in (Wang et al., 2021b), $L_2$ regularization is *increased gradually* to a very strong level, while here we use a *fixed* strong regularization, which probably is the *simplest* form of utilizing a strong regularization.

## 5    Rectified Argument on Value of Pruning

When addressing concerns from the reviewers, we find a phenomenon against our "pruning has value" argument. Specifically, in Tab. 7, we provide the scratch training results with two random

Table 8: Test accuracies (%) of pruning ResNet56 on CIFAR10. Unpruned accuracy: 93.78%. Each setting is randomly run 3 times. "Acc. gain" means the accuracy gain of initial LR $10^{-2}$ over $10^{-3}$.

| Pruning ratio $r$ | 0.5 | 0.7 | 0.9 | 0.95 |
|---|---|---|---|---|
| Sparsity/Speedup | 49.82%/1.99× | 70.57%/3.59× | 90.39%/11.41× | 95.19%/19.31× |
| | Finetuning LR schedule: 120 epochs, initial $10^{-2}$, decay 60/90 | | | |
| Train from scratch | $92.78_{\pm 0.23}$ | $92.11_{\pm 0.12}$ | $88.36_{\pm 0.20}$ | $84.60_{\pm 0.14}$ |
| $L_1$ (Li et al., 2017) | $93.51_{\pm 0.07}$ | $92.26_{\pm 0.17}$ | $88.71_{\pm 0.15}$ | $84.63_{\pm 0.28}$ |
| $L_1$ + OrthP | $93.36_{\pm 0.19}$ | $91.96_{\pm 0.06}$ | $86.01_{\pm 0.34}$ | $82.62_{\pm 0.05}$ |
| **StrongReg** | $\mathbf{93.55_{\pm 0.06}}$ | $\mathbf{92.38_{\pm 0.09}}$ | $\mathbf{89.24_{\pm 0.16}}$ | $\mathbf{85.90_{\pm 0.19}}$ |
| | Finetuning LR schedule: 120 epochs, initial $10^{-3}$, decay 80 | | | |
| $L_1$ (Li et al., 2017) | $93.12_{\pm 0.10}$ | $91.77_{\pm 0.11}$ | $87.57_{\pm 0.09}$ | $83.10_{\pm 0.12}$ |
| **StrongReg** | $\mathbf{93.44_{\pm 0.06}}$ | $\mathbf{91.96_{\pm 0.11}}$ | $\mathbf{88.69_{\pm 0.19}}$ | $\mathbf{85.45_{\pm 0.25}}$ |
| Acc. gain. ($L_1$) | 0.39 | 0.49 | 1.14 | 1.53 |
| Acc. gain. (StrongReg) | **0.11** | **0.42** | **0.55** | **0.45** |

initialization schemes, kaiming uniform and orthogonal initialization. The former is the default PyTorch initialization scheme for convolutional and linear layers[2]; the latter is proposed by Saxe et al. (2014) which can achieve exact isometry. The problem is, the best pruning results in Tab. 7 using OrthP are generally *on par with* the scratch training results using orthogonal initialization. Namely, pruning is shown no value therein. How should we respond to this observation against our prior claim and how does it affect our conclusions?

We present another set of results – We use MLP-7-ReLU as evaluation network instead of MLP-7-Linear above. Clearly, here we want to see if the *non-linearity* can make any difference. Results are shown in Tab. 9, where the dynamical isometry recovery method is changed from OrthP to StrongReg because OrthP is proposed for *linear* MLP networks while here the MLP is non-linear.

Table 9: Test accuracies (%) of $L_1$-norm pruning with **MLP-7-ReLU** network on MNIST. Accuracy of unpruned model: 98.16%. Each setting is randomly run 5 times, mean accuracy and stddev reported. Hyper-parameters: batch size 100, SGD optimizer, weight decay 0.0001, LR schedule: initial LR 0.01, decayed at epoch 30 and 60 by factor 0.1, total epochs: 90. The LR schedule is used for both scratch training and finetuning.

| Finetuning setting | Pruning ratio 80% | Pruning ratio 90% |
|---|---|---|
| LR $10^{-2}$, 90 epochs | $96.75_{\pm 0.09}$ | $94.76_{\pm 0.15}$ |
| LR $10^{-2}$, *StrongReg*, 90 epochs | $\mathbf{96.98_{\pm 0.04}}$ | $\mathbf{95.10_{\pm 0.13}}$ |
| Scratch (Kaiming uniform) | $96.60_{\pm 0.16}$ | $94.64_{\pm 0.24}$ |
| Scratch (Orthogonal init) | $96.52_{\pm 0.17}$ | $92.56_{\pm 1.90}$ |

As seen, now, either using kaiming uniform or orthogonal initialization, pruning is consistently better than scratch training by a fair margin. The advantage is amplified when using StrongReg.

Then, a worthy question here is: **Why on the MLP-7-Linear, pruning is not better than orthogonal initialization, while on MLP-7-ReLU, it *is*?**

This is actually straightforward to see if we see pruning *as a kind of initialization*. As mentioned above, on linear MLP networks, orthogonal initialization is proven to be the *optimal* in the sense of exact isometry. That is, no other initialization can be better. Pruning is essentially also *a kind of initialization* for the subsequent fine-tuning, so not surprisingly, it cannot beat the (optimal) orthogonal initialization, hence no value. However, when it comes to the MLP-7-ReLU network, orthogonal initialization is no longer optimal (as mentioned, there has been *no* method up to date that can achieve exact isometry for *non-linear* networks). Then, it is likely that pruning provides better initialization weights than orthogonal initialization or kaiming uniform. It just turns out that pruning really achieves this, and archives more with the help of StrongReg.

There are at least 3 key takeaways from the above toy but inspiring comparisons:

---
[2]https://github.com/pytorch/pytorch/blob/68d8ab0cc60536db5a9af4c08ff39e43b252802f/torch/nn/modules/linear.py#L96

- First, the comparison above shows that whether pruning has value seriously hinges on the *network type, initialization scheme, and pruning ratios (and maybe more)*. Looking at pruning as a kind of initialization as we do in this paper (and examine it through the lens of dynamical isometry) is actually a pretty good perspective, from which many results that *appear* inconsistent or random start to be explainable and logically consistent.

- Second, pruning shows the value in the non-linear case, but no value in the linear case. On the whole, we believe it is fair to say pruning *has* value considering the non-linear case is more practical.

- Third, more profoundly, the above results actually suggest, *if dynamical isometry is fully recovered, pruning will (probably) have no value indeed because it cannot beat the initialization scheme that can achieve exact isometry*. Acquired with this knowledge, we update our prior "pruning has value" argument to a more rigorous one: "**pruning has the potential to be valuable if the random initialization of scratch training cannot achieve exact isometry**". Practically, up to date, for non-linear networks (not to mention BN, residuals, convolution), there has been no such method that can achieve exact isometry. Thus, it is still *likely* for pruning to be valuable at present.

## 6 CONCLUSION

In this work, we present extensive empirical evidences to show the "no value of inheriting weights" argument in prior works is inaccurate because of improper finetuning LR schedules. We further tap into dynamical isometry to explain why the finetuning LR has such a great impact on the final performance, through carefully designed control experiments with 4 hypotheses. We show whether pruning has value seriously depends on the context (*e.g.*, network type, random initialization scheme, pruning ratio). Looking at pruning as a kind of initialization is a favorable perspective that can make seemingly inconsistent and random results become predictable and coherent. For practically amending the broken dynamical isometry, we also present a rather simple regularization-based technique that works effectively on residual convolutional networks.

The finding of dynamical isometry in structured pruning in this paper justifies the value of inheriting weights, in line with the past research wisdom and our common beliefs. It also helps us towards a better understanding of pruning and possibly can inspire more advanced pruning algorithms as dynamical isometry recovery has been shown a worthy direction in the paper. In addition, the awareness of dynamical isometry in structured pruning can help us render a more faithful conclusion when comparing different pruning methods.

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

## A    RESULTS OF REEXAMINATION OF (CROWLEY ET AL., 2018)

Table 10: Test accuracy comparison between 2 pruning schemes and 4 scratch training schemes in (Crowley et al., 2018). Network: **WRN-40-2** (unpruned accuracy: 95.08%, params: 2.24M). Dataset: **CIFAR-10**. Results above the dashline are directly cited from (Crowley et al., 2018); results below the dashline are from our reproducing (with the official code of (Crowley et al., 2018) at https://github.com/BayesWatch/pytorch-prunes for fair comparison). "Rerun" means we rerun the code of (Crowley et al., 2018) as it is. "LR $10^{-2}$" means we redo the finetuning for the pruned models in "Rerun" *using our finetuning LR schedule* (120 epochs, initial $10^{-2}$, decay 60/90). Finetuning is randomly repeated for 3 times, mean (stddev) accuracies reported. The main point here is that (Crowley et al., 2018) draws the conclusion that scratch training is better than pruning *because of an improper finetuning LR scheme*. With the proper finetuning LR scheme, pruning is actually *better* than scratch training.

| Method | Scratch? | 500K params budget | | 1M params budget | | 1.5M params budget | |
| --- | --- | --- | --- | --- | --- | --- | --- |
| | | Params (M) | Acc. (%) | Params (M) | Acc. (%) | Params (M) | Acc. (%) |
| $L_1$-norm pruning (Li et al., 2017) | ✗ | 0.51 | 90.86 | 1.02 | 92.61 | 1.52 | 93.63 |
| Fisher pruning (Theis et al., 2018) | ✗ | 0.52 | 92.59 | 1.02 | 93.51 | 1.52 | 94.51 |
| Varying Depth | ✓ | 0.69 | 93.56 | 1.08 | 94.54 | 1.47 | 94.64 |
| Varying Width | ✓ | 0.50 | 93.45 | 0.98 | 94.30 | 1.48 | 94.66 |
| Varying Bottleneck | ✓ | 0.50 | 93.69 | 1.00 | 94.40 | 1.49 | 94.79 |
| Fisher Scratch | ✓ | 0.52 | **93.72** | 1.02 | **94.65** | 1.52 | **94.86** |
| $L_1$-norm pruning (Li et al., 2017) (Rerun) | ✗ | 0.50 | 91.23 | 1.00 | 92.80 | 1.51 | 93.52 |
| $L_1$-norm pruning (Li et al., 2017) (LR $10^{-2}$) | ✗ | 0.50 | 93.88 (0.10) | 1.00 | 94.49 (0.10) | 1.51 | 94.92 (0.16) |
| Fisher pruning (Theis et al., 2018) (Rerun) | ✗ | 0.52 | 92.17 | 0.98 | 93.57 | 1.48 | 94.67 |
| Fisher pruning (Theis et al., 2018) (LR $10^{-2}$) | ✗ | 0.52 | **94.27** (0.09) | 0.98 | **94.80** (0.02) | 1.48 | **95.10** (0.13) |

## B    TRAINING SETTING SUMMARY

There are three datasets in our experiments in the paper: MNIST, CIFAR10, and ImageNet. Apart from some key settings stated in the paper, a more detailed training setting summary is shown as Tab. 11. We use 8 NVIDIA V100 GPUs for all our experiments.

Table 11: Training setting summary in finetuning. For the SGD solver, in the parentheses are the momentum and weight decay.

| Dataset | MNIST | CIFAR10 | ImageNet |
| --- | --- | --- | --- |
| Solver | SGD (0.9, 1e-4) | SGD (0.9, 5e-4) | SGD (0.9, 1e-4) |
| LR schedule (Initial 1e-2) | Multi-step (decay 30/60) | Multi-step (decay 60/90) | Multi-step (decay 30/60/75) |
| LR schedule (Initial 1e-3) | Multi-step (decay 45) | Multi-step (decay 80) | Multi-step (decay 45/68) |
| Total epoch | 90 | 120 | 90 |
| Batch size | 100 | 128 | 256 |
| Data augmentation | None | Random crop and horizontal flip | Random crop and horizontal flip |

A typical finetuning LR schedule looks like this in our paper: "90 epochs, initial $10^{-2}$, decay 30/60". Its meaning is: the total number of training epochs is 90; initial LR is $10^{-2}$; at epoch 30 and 60, LR is multiplied by a default factor 0.1. The others can be inferred likewise.

## C    WHY OUR REPRODUCED RESULTS ARE BETTER THAN (LIU ET AL., 2019)

In Tab. 1, we present the results of ResNet34 on ImageNet, pruned by the $L_1$-norm pruning method (Li et al., 2017). Readers may be curious that *why our scratch-training results are much*

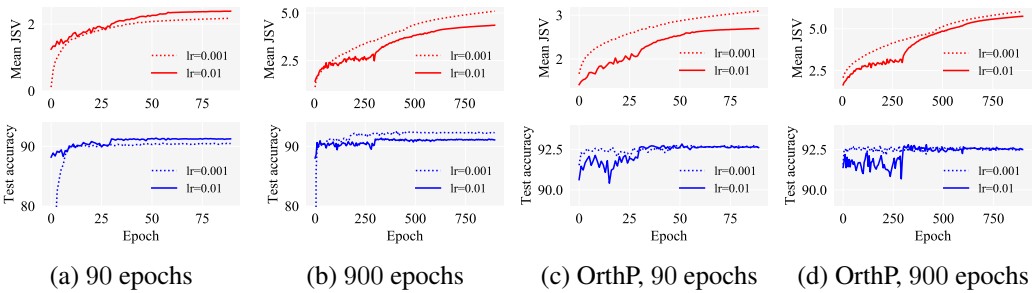

Figure 2: Mean JSV and test accuracy during finetuning with different setups. Note that with OrthP (c, d), mean JSV recovers faster; so does the test accuracy. The pruning ratio in this case is 0.9. (*This figure is best viewed in color*).

*higher than those of Rethinking (Liu et al., 2019) (by* $0.4$-$0.6\%$). This has a historical reason. (Liu et al., 2019)[3] refers to the official PyTorch ImageNet example[4], where the LR is only decayed *twice* (from 1e-1 to 1e-2, then to 1e-3). However, by our empirical observation, 1e-3 is still not the lowest LR that the network converges. If the LR is decayed another time (to 1e-4), the top-1 accuracy can still bump by round 0.5-0.8% point, which is a significant improvement for ImageNet thus not negligible. Therefore, we choose to decay the LR further to 1e-4 and finally to 1e-5 to ensure the network is *fully converged*. This reiterates the comparison rule of our paper: comparing different methods *in their best shape*. Comparing scratch training to $L_1$-norm pruning *before* the finetuned network finally converges may appear fair (given the same number of epochs) but is of less practical meaning and may hide the true picture.

## D    HOW TO *Properly* LOOK AT THE MEAN JSV METRIC FOR DYNAMICAL ISOMETRY

In the main paper, we use mean JSV as the metric to measure dynamical isometry, for the following 3 specific reasons: (1) It was used by Lee et al. (2020), which analyzes the dynamical isometry for randomly initialized network. Given its success there, it is very natural for us to also employ this metric for analyzing pretrained networks here. (2) We currently do not have a better alternative, either. (3) In practice, we find mean JSV is informative (*e.g.*, in Tab. 7, the mean JSV trend is well-correlated with the test accuracy trend), as long as we see it *properly*.

This section is meant to provide more background regarding how to look at the mean JSV for dynamical isometry *properly*. It is not a new invention of our paper but a general practical guideline about the relationship between mean JSV and dynamical isometry in order to help readers better understand our paper.

DI (dynamical isometry) is defined by mean JSV close to 1 in Saxe et al. (2014). Rigorously, in Saxe et al. (2014), DI describes the distribution of *all* JSVs. Mean JSV is only an average sketch of the distribution. Nevertheless, this average approximation is accurate enough for analysis. In other words, *if a network has mean JSV close to 1, we can say this network has dynamical isometry*.

Then, a non-trivial technical question is: **When we deal with practical DNNs in the real world, how close is the so-called "close to 1"?** To our best knowledge, there is no outstanding theory to quantify this, so we resort to empirical analysis, specifically on the MLP-7-Linear network used in the main paper.

As seen, there is a clear trend in Tab. 12: *larger* pruning ratio, *smaller* mean JSV, and *lower* test accuracy (either before or after finetuning).

Particularly note the mean JSV range where the pruned network can be *finetuned back to the original accuracy (*92.77%*)*, which is $>= 0.0151$. This means, for networks with mean JSV greater than

---

[3]https://github.com/Eric-mingjie/rethinking-network-pruning/tree/master/imagenet/l1-norm-pruning
[4]https://github.com/pytorch/examples/tree/master/imagenet

Table 12: Mean JSV and test accuracies (%) of MLP-7-Linear on MNIST under different pruning ratios. Each result is randomly run for 3 times. We report the mean accuracy and (std). "ft." is short for finetuning.

| Pruning ratio | 0 | 0.1 | 0.2 | 0.3 | 0.4 | 0.5 | 0.6 | 0.7 | 0.8 | 0.9 |
|---|---|---|---|---|---|---|---|---|---|---|
| mean JSV | 2.4987 | 1.7132 | 0.9993 | 0.5325 | 0.2711 | 0.1180 | 0.0452 | 0.0151 | 0.0040 | 0.0004 |
| Acc. before ft. | 92.77 | 91.35 | 78.88 | 62.21 | 32.14 | 11.47 | 9.74 | 9.74 | 9.74 | 9.74 |
| Acc. after ft. | / | 92.82 (0.05) | 92.80 (0.04) | 92.80 (0.01) | 92.77 (0.01) | 92.77 (0.02) | 92.77 (0.00) | 92.78 (0.02) | 91.37 (0.03) | 87.82 (0.03) |

0.0151, in spite that their immediate accuracies (without finetuning) can be distinct (*e.g.*, 91.35% at PR 0.1 vs. 9.74% at PR 0.7), intrinsically, they are *equivalently potential* after finetuning.

DI theory suggests that mean JSV equal to 1 is the best case. Then we would ask, how about mean JSV equal to 2? 1.5? 0.8? Are they really worse than the ideal value 1? Tab. 12 above shows *not necessarily*, because a network with mean JSV 0.2711 can reach comparable accuracy to the network with mean JSV 1. Only when the mean JSV is smaller than some threshold (in this case, 0.0151), it leads to an irrecoverable damage to the network optimization, which eventually results in lower generalization ability (*i.e.*, test accuracy).

By this "trainable with equivalent potential" rule, **if a mean JSV lies in the range of $>= 0.0151$, we can regard it as "close to 1"** because they can do *just as well as* 1.

Here, we only mention the *lower* bound, how about the *upper* bound? Can mean JSV $10,000$ also be regarded as "close to 1"? This is a good question worth further dedicated investigation. Yet, for now, we do not need to worry much about it, because in practice, a *normally* trained network *rarely* presents a very large mean JSV by our empirical observation, but is quite likely to have a very small mean JSV (0.0004 at pruning ratio 0.9 in Tab. 12 is a concrete example).

## E   HOW STRONGREG IS RELATED TO DYNAMICAL ISOMETRY

**The intuition behind StrongReg**. Dynamical isometry describes a nice state of the network that signals can propagate through it without serious magnitude explosion or attenuation. Since the weights in a network are dependent on each other, removing some of them will definitely hurt the isometry because it is based upon all the weights. When we use a strong regularization to push these unimportant parameters to zero, it explicitly makes the other parameters learn to not rely on them, that is, encouraging the gradients not to pass through these weights/neurons because they are going to be cleaned out. This way, when the unimportant parameters are physically removed, it will incur much less damage to the left parameters, thus maintaining the network dynamical isometry well.

**StrongReg works by improving dynamical isometry**. Here we apply StrongReg to pruning the MLP-7-Linear network at pruning ratio 0.8 and 0.9.

Table 13: Test accuracies (%) of applying StrongReg to pruning **MLP-7-Linear** network on MNIST. Accuracy of unpruned model: 92.77%. Each setting is randomly run 5 times, mean accuracy and stddev reported. "Acc. gain" refers to the mean accuracy improvement of LR $10^{-2}$ over $10^{-3}$. Hyper-parameters: batch size 100, SGD optimizer, weight decay 0.0001, LR schedule: initial LR 0.01, decayed at epoch 30 and 60 by factor 0.1, total epochs: 90. The LR schedule is used for both scratch training and finetuning.

| Finetuning setting | Pruning ratio 80% | | | Pruning ratio 90% | | |
|---|---|---|---|---|---|---|
| | LR $10^{-3}$ | LR $10^{-2}$ | Acc. gain | LR $10^{-3}$ | LR $10^{-2}$ | Acc. gain |
| 90 epochs | $90.54_{\pm 0.02}$ | $\mathbf{91.36_{\pm 0.02}}$ | 0.82 | $87.59_{\pm 0.01}$ | $\mathbf{87.81_{\pm 0.03}}$ | 0.22 |
| OrthP, 90 epochs | $92.77_{\pm 0.03}$ | $\mathbf{92.79_{\pm 0.03}}$ | 0.02 | $92.72_{\pm 0.03}$ | $\mathbf{92.77_{\pm 0.04}}$ | 0.05 |
| StrongReg, 90 epochs | $\mathbf{92.80_{\pm 0.04}}$ | $\mathbf{92.80_{\pm 0.02}}$ | 0.00 | $92.48_{\pm 0.02}$ | $\mathbf{92.52_{\pm 0.04}}$ | 0.04 |

As seen in Tab. 13, (1) similar to OrthP, StrongReg can rectify the test accuracy from the under-rated ones to $92.48 \sim 92.80$, close to the best possible performance (around 92.77); (2) StrongReg can close the performance gap between LR $10^{-3}$ and $10^{-2}$, just like OrthP. In short, in terms of performance, StrongReg behaves very similarly to OrthP.

To further show StrongReg really works *by maintaining dynamical isometry*, like Tab. 4 and Tab. 5 above, we list the first 10-epoch mean JSV and test accuracies (at PR 0.9) and get the following summarized results:

Table 14: Mean JSV of the first 10 epochs under different finetuning settings. Epoch 0 refers to the model just pruned, before any finetuning. Pruning ratio is 0.9. Note, with OrthP, the mean JSV is 1 because OrthP can achieve exact isometry.

| Epoch | 0 | 1 | 2 | 3 | 4 | 5 | 6 | 7 | 8 | 9 | 10 |
|---|---|---|---|---|---|---|---|---|---|---|---|
| LR=$10^{-2}$, w/o OrthP | 0.0004 | 0.6557 | 0.8946 | 1.0191 | 0.9826 | 1.0965 | 1.1253 | 1.2595 | 1.3298 | 1.2940 | 1.4238 |
| LR=$10^{-3}$, w/o OrthP | 0.0004 | 0.0004 | 0.0006 | 0.0014 | 0.1103 | 0.2765 | 0.3501 | 0.4320 | 0.5167 | 0.7478 | 0.8501 |
| LR=$10^{-2}$, w/ OrthP | 1.0000 | 1.2318 | 1.4144 | 1.4277 | 1.4017 | 1.4709 | 1.5171 | 1.5551 | 1.6082 | 1.6538 | 1.6648 |
| LR=$10^{-3}$, w/ OrthP | 1.0000 | 1.5135 | 1.6630 | 1.7449 | 1.8250 | 1.8720 | 1.9193 | 1.9556 | 1.9943 | 2.0084 | 2.0409 |
| LR=$10^{-2}$, w/ StrongReg | 3.0275 | 2.1538 | 2.0390 | 1.0191 | 1.9696 | 2.1011 | 1.9734 | 2.0810 | 2.0541 | 2.0563 | 2.0581 |
| LR=$10^{-3}$, w/ StrongReg | 3.0275 | 2.9691 | 2.9514 | 2.9921 | 2.9869 | 3.0101 | 3.0448 | 3.0516 | 3.0555 | 3.0441 | 3.0244 |

Table 15: Test accuracy (%) of the first 10 epochs *corresponding to Tab. 14* under different finetuning settings. Epoch 0 refers to the model just pruned, before any finetuning. Pruning ratio is 0.9.

| Epoch | 0 | 1 | 2 | 3 | 4 | 5 | 6 | 7 | 8 | 9 | 10 |
|---|---|---|---|---|---|---|---|---|---|---|---|
| LR=$10^{-2}$, w/o OrthP | 9.74 | 63.86 | 79.96 | 79.74 | 80.06 | 85.79 | 85.82 | 86.11 | 86.45 | 86.53 | 85.95 |
| LR=$10^{-3}$, w/o OrthP | 9.74 | 9.74 | 9.74 | 12.09 | 21.74 | 27.95 | 33.55 | 35.92 | 49.19 | 65.50 | 69.90 |
| LR=$10^{-2}$, w/ OrthP | 9.74 | 91.05 | 91.39 | 91.33 | 91.37 | 91.74 | 91.69 | 90.74 | 91.39 | 91.58 | 91.44 |
| LR=$10^{-3}$, w/ OrthP | 9.74 | 90.81 | 91.59 | 91.77 | 91.85 | 92.04 | 92.12 | 92.22 | 92.12 | 92.33 | 92.25 |
| LR=$10^{-2}$, w/ StrongReg | 89.13 | 91.68 | 91.11 | 91.37 | 91.84 | 91.47 | 90.11 | 91.32 | 90.78 | 91.55 | 91.29 |
| LR=$10^{-3}$, w/ StrongReg | 89.13 | 92.03 | 91.84 | 92.13 | 92.05 | 91.85 | 92.00 | 92.02 | 92.09 | 92.08 | 92.08 |

(1) First, in Tab. 14, at epoch 0, the mean JSV is around 3 (which can be considered close to the exact isometry 1, by our analysis in Sec. D) vs. 0.0004 when the dynamical isometry is not recovered. (2) Second, the test accuracy also has a high starting point: at epoch 1, using StrongReg achieves 91.68 (LR $10^{-2}$) and 92.03 (LR $10^{-3}$) vs. 63.86 (LR $10^{-2}$) and 9.74 (LR $10^{-3}$) when the dynamical isometry is not recovered.

Particularly note StrongReg can achieve similar mean JSV and test accuracy to OrthP, showing that StrongReg works by maintaining dynamical isometry, as OrthP does.

