# OpenReview forum: "Rethinking Again the Value of Network Pruning -- A Dynamical Isometry Perspective"
_ICLR.cc/2022/Conference — ICLR 2022 Submitted_

### Official Review · Reviewer_YYie · 2021-11-01

**Correctness:** 2
**Technical Novelty And Significance:** 2
**Empirical Novelty And Significance:** 2
**Recommendation:** 5
**Confidence:** 4

**Main Review:**

1) **Nice analysis of previous papers**: It is commendable that this paper points out issues with two previous papers (Crowley et al 2018, Liu et al 2019) who make methodological errors by not tuning the learning rate for fine-tuning, and use the results of these erroneous experiments to make general statements about the efficacy of pruning (regardless of whether these statements are true or not). However from the perspective of these previous papers, it seems they had a fixed small fine-tuning budget of 20 epochs, and did not want to exceed this as we require number of fine-tuning epochs to be much smaller than the number of training epochs. While I do think that this requirement is arbitrary, it helps understand the claims of the previous papers.

2) **Unfair experimental comparison**: In Table 1, it seems that the small Resnet models (A,B) trained from scratch are trained for 120 epochs, while the best prune-finetuned models are trained for 90 epochs. However also accounting for the training of the original large Resnet-34 model, the total number of training epochs are 90+90=180 epochs. Hence it seems that the prune+finetuned models are trained overall for a larger number of epochs, which may be a confounding factor which explains their better results. I would *strongly* suggest that the experiments be run such that the number of training epochs for "training small model from scratch" and "large model training with pruning and finetuning" be equalized for a fair comparison.

3) **Unclear why hypothesis must be true**: This paper rest on the hypothesis that training a large model followed by pruning and fine-tuning offers improved generalization benefits over training a small model from scratch. In principle it should be clear that both two methods can produce models with similar generalization given sufficiently strong optimizers, as the only difference between them is the initialization of the small model. Even in practice, we often are able to train models such that the train loss is close to zero (or sufficiently small), regardless of whether we train small models from scratch or fine-tune pruned models.
Hence given that both methods solve the same overall optimization problem (i.e, minimizing loss of the small model), I am confused regarding why one would expect pruning + fine-tuning to generalize better than training from scratch, even in principle. For example, it is clear that given infinite epochs and a suitable (well-tuned) optimizer both methods should work equally well. Is the paper's hypothesis a claim about the speed of optimization, that pruning+fine-tuning is a fast way to obtain a small model? If so, how does this change after having accounted for the training of the dense model itself (which is usually slower to train)?

4) **Incorrect metric for measuring Dynamical Isometry**: In equation 1, the paper uses mean singular values as a metric for assessing dynamical isometry. However if we require that all singular values are equal to one, then the metric to use would be the sum of (squared) deviations from unity of each singular value (\sum_i (\Sigma_{ii} - 1)^2). In such a metric, zero corresponds to dynamical isometry being achieved. However the metric of mean singular values has no such meaning as the standard deviation can be large. I am hence confused regarding why the paper chose to use the mean JSV metric, and if there is no strong reason for doing so, I would *strongly* recommend re-running experiments with the deviations-from-unity metric.

5) **Relationship between mean JSVs and convergence unclear**: Hypotheses 1-4 and the boxed explanation talk about the "recovery" of mean JSVs and their relationship to network convergence. I am confused regarding why the behaviour of mean JSVs during or after training is deemed meaningful. I understand that too small or too large mean JSVs must be avoided at initialization as they can cause vanishing or exploding gradients, but can increasing mean JSVs from ~1 to ~5 (as in Fig 2) be deemed meaningful? It is thus not clear why mean JSV (or any measure of dynamical isometry) can be used to measure model convergence (ref. to boxed explanation in page 7). For example, if gradient-norm regularization is used, then the model would converge and yet have small JSVs. The only measure of model convergence is the gradient norm of the loss w.r.t. weights, and other measures are meaningless unless explicitly shown otherwise.

**Summary Of The Paper:**

This work takes a second look at the set of papers (Crowley et al 2018, Liu et al 2019) which claim that there is no generalization benefit offered by pruning, i.e., training a large network first and then pruning performs identically or worse than training the same pruned network architecture from scratch. This work claims that these previous works do not set learning rates during fine-tuning correctly while performing their experiments and hence these claims are invalid. The paper also offers an explanation for this phenomenon via dynamical isometery theory.

**Summary Of The Review:**

This paper does not make a solid case for the hypothesis that "pruning+finetuning produces models with better generalization than training small models from scratch", as there are issues with both the experimental and the analytical studies. In the experimental setup, the comparisons are not performed fairly, while in the analytical parts use flawed metric to measure model convergence and dynamical isometry. Even fundamentally, it is unclear why the proposed hypothesis must be true in the first place. For all these reasons, I would recommend rejection.

Post rebuttal update: The discussions did not change my opinion of the paper. However I appreciate the efforts taken by the authors to update the paper in light of reviewer suggestions, particularly to make the core argument more rigorous. Although I do not think the paper still succeeds at this, I am increasing my score slightly to reflect this change.

---

> ### Author Response · Authors · 2021-11-21
> **Responses to Reviewer YYie (R4) -- Part 3**
>
> `R4-Weakness4`: Relationship between mean JSVs and convergence unclear.
>
> `R4-Response4`:
> We use mean JSV because of the 3 reasons we explain in `R4-Response3`. We also think *increasing mean JSVs from ~1 to ~5 (as in Fig 2)* cannot be deemed meaningful. The point of Fig. 2 is in the early training stages. This figure is not well-plotted, so we have shown it in Tab. #2, #3 (see `[Response to All Reviewers] How to properly look at the mean JSV metric for dynamical isometry`), which should be more straightforward.
>
> In principle, mean JSV does not imply convergence itself, just implying the *easiness of convergence*. We actually can intentionally set it to any positive number we want (just like above in `R4-Response3`, we can set it to 3 manually). But for a *normally* trained network, we do find mean JSV correlates with convergence or generalization especially for the early training stages as we show in `[Response to All Reviewers] How to properly look at the mean JSV metric for dynamical isometry`.
>
> A good metric indeed needs more deliberation. Right now, we do not have a better alternative, and considering it is still fairly useful and used by previous works, we hope R4 can see our efforts here. We are definitely more than happy to include better metrics (if we find them) to re-evaluate our hypotheses.
>
> Hope our response can help clear your concerns. **If you have further questions regarding our feedback, just let us know!**

---

> > ### Comment · Reviewer_YYie · 2021-11-28
> > **Response**
> >
> > `Weakness 1` I don't think practicality is an issue here, as we spend as much number of epochs pruning as we do training from scratch. For example, as in this paper, models are pruned for 90 epochs and the large model training itself takes 90 epochs. In any case, it is critical that the paper make the precise statement it intends to show or disprove. There is no mention of such practicality constraints in the paper as far as I can see. The hypothesis mentioned in section 2.1 is that "Inheriting weights from a pretrained model in pruning has no value, i.e., training from scratch the small model can match (or outperform sometimes) the counterpart pruned from a big pretrained model", which makes no statements regarding practicality.
> >
> > It is also possible that the authors have misunderstood my argument. When 5 models are pruned in parallel, we would still need to compare against 180 epochs trained-from-scratch model as each such pruned model is only trained for 180 epochs each. The point is to equalize the number of optimizer iterations that these models have seen.
> >
> > `Weakness 2` I think the authors have misunderstood my point yet again. My point is that even if pruned models outperform models trained from scratch, under what conditions must we expect that to happen? For instance, if dynamical isometery is achieved, and one trains a model from scratch long enough, one would expect that pruned models work as well as models trained from scratch according to the ideal conditions. Again, I think one way to be clear about these is to **clearly** and **rigorously** state the hypothesis that the paper intends to prove or disprove, as claims regarding such optimizer practicality were never brought up in the original paper. Without this, one can very easily write endless follow up papers "proving" and "disproving" the same hypothesis. A clear hypothesis in your case would roughly state that "under practical conditions such as blah blah, we expect pruning for x epochs to outperform training from scratch done for y epochs".
> >
> > `Weakness 3` The argument that deviation-from-unity is not better than mean JSV is confusing. If dynamical isometery measures how close each of the spectral values are equal to one, then it is clear that deviation-from-unity measures **exactly** this phenomenon, which mean JSV misses. The authors bring up auxiliary issues that are thought to be related to the **outcomes** of dynamical isometery such as model performance and trainability, which is not the point at all. Nonetheless it is nice to see results with deviation-from-unity as well.
> >
> > `Weakness 4` Regarding mean JSV and convergence, my point is that there is no underlying theory connecting both concepts. The paper claims some correlations between both, but this means that high mean JSV need not imply good convergence and so on.
> >
> > > A good metric indeed needs more deliberation. Right now, we do not have a better alternative
> >
> > There is indeed a perfectly good way to detect convergence, which I also mentioned earlier, which is to look at the loss gradients w.r.t. parameters, which must be equal to zero.

---

> > > ### Author Response · Authors · 2021-11-28
> > > **Thanks for Your Further Comments!**
> > >
> > > `Weakness1`: *"There is no mention of such practicality constraints in the paper as far as I can see"* -- What we initially reported is the most typical pruning comparison setting. Since R4 mentioned the 180-epoch comparison setup in the first place, we point out this is *not* the typical way in the pruning community, hence the emphasis on practicality. The practicality consideration, (we thought), is obvious to see given pruning is a topic for improving DNN efficiency in real-world applications. Thanks for letting us know that this actually is not so obvious in our paper right now. We'll explicitly state we focus on the practicality of pruning algorithms and the "value of pruning" argument is treated under this practical context.
> > >
> > > "*When 5 models are pruned in parallel, we would still need to compare against 180 epochs trained-from-scratch model as each such pruned model is only trained for 180 epochs each.*" -- We think we understand this point correctly. This is why we mentioned if using the scratch training scheme suggested by R4, training 5 pruned models from scratch takes 180x5=900 epochs in total.
> > >
> > > "*The point is to equalize the number of optimizer iterations that these models have seen.*" -- We understand this and fully agree with it actually. Yet, this is exactly where we mentioned the necessity of considering practicality. We do not think this is the "point" (at least for this paper), because even under this setting scratch training is shown no worse than pruning, it answers little to the argument of "value of pruning" *in practice*. Again, we still see our paper as one focusing on the *practical* value of pruning in the real world.  This is why we keep emphasizing what we reported is the most practical or typical comparison setting in the community right now, also the typical way of pruning being applied in industry.
> > >
> > > `Weakness 2`: Thanks for this point! We fully agree with it. Even during this rebuttal, we also feel we are getting more precise about what should be rigorously stated. This is why we added the section ''Rectified Argument on Value of Pruning" to make our statements more rigorous.
> > >
> > > Meanwhile, we hope R4 can see that compared to the previous "rethinking the value of pruning" papers (Liu et al., 2019) (Crowley et al., 2018), which are purely empirical studies, our work has presented a more principled framework via dynamical isometry to make the "value of pruning" argument *more rigorous* already. We believe this can be fairly regarded as a non-trivial contribution. R1 also agrees with this point: *"Though some issues exists and DI may not be the ultimate solution to the question, `the paper provides valuable thoughts for this filed`"*. So we sincerely hope R4 can see the value of this paper and kindly improve your score slightly even though you think it is not good enough for publication.
> > >
> > > `Weakness 3`: "*If dynamical isometry measures how close each of the spectral values are equal to one, then it is clear that deviation-from-unity measures exactly this phenomenon, which mean JSV misses.*" -- We also fully agree with this point. Again, this is where we emphasize practicality. Dynamical isometry is studied for the *trainability/initialization* of deep networks, where researchers have the freedom to decide what should be the best value of mean JSV (which is 1, obviously). However, when the network starts training, no one can ensure the mean JSVs always keep at 1 during training. R4's statement "it is clear that deviation-from-unity measures exactly this phenomenon" should be "it is clear that deviation-from-unity measures exactly this phenomenon *at initialization*", while our paper discusses the mean JSV  behaviors *during training*. This is also why the deviation-from-unity metric does not work well for our case, as we've shown.
> > >
> > > `Weakness 4`: "*there is no underlying theory connecting both concepts*" -- We fully agree that mean JSV has *no* relation with convergence (what we said is "mean JSV implies the *easiness* of convergence", not convergence per se). This is also why we report the *first 10-epoch* mean JSV and accuracies (in merely 10 epochs, the model is not converged at all). R4 may misread our responses.
> > >
> > > "*There is indeed a perfectly good way to detect convergence, which I also mentioned earlier, which is to look at the loss gradients w.r.t. parameters, which must be equal to zero.*" -- Thanks for bringing this up again. But notably, we are not needing a tool to detect *convergence* (even by looking at the learning curves, we can easily see if a model is converged or not). We need a tool to detect the *easiness* of convergence, which implies the potential of a model given N-epoch (like 90-epoch) finetuning. This is actually what mean JSVs faithfully describe, as we've shown in `How to properly look at the mean JSV metric for dynamical isometry` (better mean JSV, fewer epochs to reach accuracy like 92%) .

---

> > > > ### Comment · Reviewer_YYie · 2021-11-30
> > > > **Thank you for the updates**
> > > >
> > > > I appreciate the effort taken by the authors to update the paper to incorporate reviewer comments, and I will increase my score accordingly. However I still do not feel the paper makes rigorous arguments and experiments to back its core argument regarding the effectiveness of pruning, I still cannot recommend acceptance.

---

> > > > > ### Author Response · Authors · 2021-11-30
> > > > > **Thanks for Your Feedback**
> > > > >
> > > > > We greatly thank R4 for your kind words and for improving the score.
> > > > >
> > > > > Regarding "*I still do not feel the paper makes rigorous arguments and experiments to back its core argument regarding the effectiveness of pruning*", we feel we may need to make a little further remarks. Note, this should *not* be taken as disagreeing with R4's opinion, just sharing a few of our thoughts.
> > > > >
> > > > > We are desperate for "rigorous" (this is why when we have the MLP-7-Linear scratch results requested by R3, we immediately realize our claim should be updated further. Our efforts in the empirical study concerning how to define "close to 1" for dynamical isometry can be another example. To our best knowledge, no previous works have attempted to do so), but for any scientific exploration, there is a process to be rigorous step by step. Right now, many concepts are not clear enough, which we admit, and these are the open problems in the community. Yet, compared to previous works in this line (the two "rethinking the value of pruning" papers (Liu et al., 2019) (Crowley et al., 2018)), this work has become much more rigorous (simply put -- *Yes, it is not perfect, but getting better*). We hope this contribution can be fairly recognized.
> > > > >
> > > > > "*Lack experiments to back its core argument regarding the effectiveness of pruning*" -- Up to now, we already know we cannot really make any reliable comment regarding whether pruning has value unless we know what is the specific network/initialization scheme/pruning ratio etc. Yet, practically, the most straightforward way to evaluate this argument is to run experiments on large-scale dataset with modern networks. In this regard, even in our original paper, we have reported *extensive* experiments on ImageNet (Tab. 1, Tab. 2) with different network archs (residual network: ResNet18/34, and single-branch network: VGG11), across a large spectrum of pruning ratios (0.1 to 0.95). Results: **17/20 experiments** are against the argument in (Liu et al., 2019) (Crowley et al., 2018)). We believe this is pretty strong to support our "pruning has value" claim (note, logically, even one single counter-example is enough for refuting "pruning has no value"; right now, we have 17/20).
> > > > >
> > > > > Finally, this is what we responded to R2, which we think is also pertinent here: *the more important may not be to have a "one-sentence-fit-all" conclusion about the value of pruning (we cannot either, actually, as we've shown), but to know what is the reason under the hood answering for the mixed and seemingly inconsistent results*.
> > > > >
> > > > > Sincerely,
> > > > >
> > > > > Authors

---

> ### Author Response · Authors · 2021-11-21
> **Responses to Reviewer YYie (R4) -- Part 2**
>
> `R4-Weakness3`: Incorrect metric for measuring Dynamical Isometry.
>
> `R4-Response3`: "why the paper chose to use the mean JSV metric?" -- We have the following 3 specific reasons for using the mean JSV as metric:
> * (1) It was used by a previous related work (Lee et al., 2020) (the differences between our paper and theirs are discussed in our related work section) and presented as an informative metric tool.
> * (2) We actually do not have a better metric in our mind either.
> * (3) In Tab. #2, #3 above (see `[Response to All Reviewers] How to properly look at the mean JSV metric for dynamical isometry`), the mean JSV correlates very well with the test accuracy in the early training stages.
>
> As for the new metric (mean deviation-from-unity) suggested by R4, first, we exlain why this new metric may not be better than mean JSV. Second, as suggested, we rerun the experiments using this new metric.
>
> * First, consider this case: we have two initialization schemes for a neural network, *A: mean JSV = 0* (this can be done by removing one internal layer; since the network is disconnected, its mean JSV is definitely 0), *B: mean JSV = 3* (this can be done by re-scaling the weights in each layer by $3^{1/7}$ after using the orthogonal initialization, for the MLP-7-Linear). Then the former has mean deviation-from-unity of 1, the latter has mean deviation-from-unity of 4. By the metric, the former should be better, but clearly it is *not* since the network is not trainable at all while the latter is trainable -- we empirically confirmed this with pytorch on the MLP-7-Linear network. This is why we think the metric may not be better than mean JSV.
> * Second, we rerun the experiments using this new metric. Results on the MLP-7-Linear at PR 0.9 are reported below:
>
>
> **Tab. R4-#1. *Mean deviation-from-unity* of the first 10 epochs with different settings. Epoch 0 refers to the model just pruned, before any finetuning**
>
> | Epoch 	| 0 	| 1 	| 2 	| 3 	| 4 	| 5 	| 6 	| 7 	| 8 	| 9 	| 10 	|
> |---	|---	|---	|---	|---	|---	|---	|---	|---	|---	|---	|---	|
> | LR=0.01, w/ OrthP 	| 0 	| 0.3917 	| 0.4850 	| 0.6278 	| 0.7665 	| 0.7573 	| 1.1150 	| 1.3016 	| 1.0021 	| 1.0641 	| 1.0292 	|
> | LR=0.001,  w/ OrthP 	| 0 	| 0.6068 	| 0.8588 	| 1.0114 	| 1.1519 	| 1.3061 	| 1.4082 	| 1.4665 	| 1.5470 	| 1.6939 	| 1.7711 	|
> | LR=0.01, w/o OrthP 	| 0.9992 	| 1.6228 	| 2.0774 	| 2.5022 	| 2.8385 	| 2.4686 	| 2.7849 	| 2.8000 	| 2.7692 	| 2.8321 	| 3.2931 	|
> | LR=0.001, w/o OrthP 	| 0.9992 	| 0.9992 	| 0.9988 	| 0.9972 	| 0.9000 	| 1.1856 	| 1.4871 	| 1.9261 	| 2.4667 	| 2.8044 	| 3.1260 	|
>
> **Tab. R4-#2. Test accuracy of the first 10 epochs with different settings. Epoch 0 refers to the model just pruned, before any finetuning**
>
> | Epoch 	| 0 	| 1 	| 2 	| 3 	| 4 	| 5 	| 6 	| 7 	| 8 	| 9 	| 10 	|
> |---	|---	|---	|---	|---	|---	|---	|---	|---	|---	|---	|---	|
> | LR=0.01, w/ OrthP 	| 9.74 	| 91.05 	| 91.39 	|91.33 	|91.37 	|91.74 	| 91.69 	| 90.74 	| 91.39 	|91.58 	|91.44 	|
> | LR=0.001,  w/ OrthP 	| 9.74 	| 90.81 	| 91.59 	|91.77 	|91.85 	|92.04 	|92.12 	|92.22 	|92.12 	|92.33 	|92.25 	|
> | LR=0.01, w/o OrthP 	| 9.74 	| 63.86 	| 79.96 	| 79.74 	| 80.06 	|85.79 	|85.82 	|86.11 	|86.45 	|86.53 	| 85.95 	|
> | LR=0.001, w/o OrthP 	| 9.74 	| 9.74 	| 9.74 	| 12.09 	|21.74 	|27.95 	|33.55 	|35.92 	|49.19 	|65.50 	|69.90 	|
>
> As seen in Tab. R4-#1, (1) the mean deviation-from-unity of using OrthP is smaller than not using, which is good since we know OrthP is useful; (2) but the problem is, the new metric does not align with the test accuracy. E.g., for each row in Tab. R4-#1, the mean deviation-from-unity is arising in general, which means bad, but the test accuracy is also arising, which means good. They are contradicting with each other. And particularly note the "LR=0.001, w/o OrthP" row, at Epoch 1~5, its test accuracy is very low. This correlates well with the low mean JSV metric (see Tab. #2 above);  but using mean deviation-from-unity, the metric value stays low at around 1, which should be regarded as good, but bad in reality.
>
> Given above, the mean deviation-from-unity metric indeed has its merit, yet it may not be better than the mean JSV metric.

---

> ### Author Response · Authors · 2021-11-21
> **Responses to Reviewer YYie (R4) -- Part 1**
>
> Thank Reviewer YYie (R4) for your very helpful comments and the generous credit to our nice analysis of previous papers! We address your concerns as follows.
>
>
> `R4-Weakness1`: Unfair experimental comparison.
>
> `R4-Response1`: What R4 suggests about the scratch training epochs has a point, yet with all due respect, we might not fully agree with it in terms of the "fairness". The reasons are:
> * The pretrained models are readily available in most practical applications. In fact, the task of pruning is mostly discussed in the context of *model compression*, where a large pretrained model typically exists already. We value the fairness in theory too but as a method for practical benefits, we should also consider the real application context.
>
> * Moreover, the large model is typically trained *once* and can be pruned into *multiple* small networks. For example, if we desire 5 efficient models with different sizes for devices with different resource budgets, then obtaining all the 5 efficient models using pruning needs $90+90\times5=540$ epochs. In contrast, with the scratch scheme suggested by R4, it will need $180\times5=900$ epochs. Then this would be unfair to pruning. There is no obvious reason that the "fairness" must be defined in the one-pruned-model case rather than the multiple-pruned-model case, especially when the latter is actually more practical and more useful.
>
> * We actually have done pretty much to maintain fairness. For example, the typical scratch results are obtained through the 90-epoch ImageNet training strategy (e.g., (Liu et al., 2019) uses this strategy). We particularly note this strategy is sub-optimal so upgrade it to the 120-epoch strategy. It improves the top-1 accuracy by 0.5~1% on average (which is a non-trivial improvement on ImageNet) against the 90-epoch scheme. The value of pruning is shown compared to these stronger baselines.
>
> `R4-Weakness2`: Unclear why hypothesis must be true.
>
> `R4-Response2`: We never claimed the proposed hypothesis *must* be true. We also understand a rigorous hypothesis needs particular deliberation and extensive empirical validation, so we are very careful with our wording in the paper. We actually only said the hypothesis is *plausible* (see "This inspires us to the following plausible explanation..." on page 7). By "plausible", we mean it makes some seemingly contradictory or random results (e.g., on the MLP-7-Linear, finetuning for 90 epochs, LR 0.01 is better than LR 0.001, while finetuning for 900 epochs, the inequation is turned over) now logically consistent and predictable.
>
> R4 states: "In principle it should be clear that both two methods can produce models with similar generalization given *sufficiently strong optimizers*", "given that both methods solve the same overall optimization problem (i.e, minimizing loss of the small model), I am confused regarding why one would expect pruning + fine-tuning to generalize better than training from scratch, even in principle. For example, it is clear that given *infinite epochs and a suitable (well-tuned) optimizer* both methods should work equally well".
>
> We actually agree with R4 on all these "in principle" statements, yet we may want to emphasize we are dealing with *practical* systems. In practice, we may not have "sufficiently strong optimizers", "infinite epochs", etc. These conditions are attractive but they are non-trivial to have in practice. Specifically,
>
> * Solving "the same overall optimization problem" does not mean the two methods definitely reach similar performance in practice. E.g., for the same network training, different initialization schemes can lead to *starkly different* results (we particularly mention "initialization" as an example because pruning is essentially a way of initialization compared to scratch training). One concrete example is in Tab. R3-#2 and Tab. R3-#3 (see our response to R3 `R3-Response5`) above, kaiming_uniform and orthogonal initialization can lead to very different results (and thus conclusions). Note, kaiming_uniform is actually the default initialization scheme in pytorch. If we are not aware of the dynamical isometry issue discussed by our paper, we probably do *not even realize* it is a problem here, then we won't even try to solve it at all (via the longer training epochs, different initializations or stronger optimizers etc., even if these are practically available).
>
> * We agree with R4, in the ideal case (when the exact isometry is achieved), pruning and scratch training should lead to similar performance (you may see our response to R3 `R3-Response5`). However, for *practical* modern deep networks, exact isometry in initialization has not been achieved up to date. Then pruning still has a chance to be valuable.

---

> ### Author Response · Authors · 2021-11-27
> **Sincerely expecting further discussions with Reviewer YYie (R4)**
>
> Dear Reviewer YYie,
>
> We greatly thank you for the reviewing process so far! Given the ICLR final discussion deadline (11/29) is approaching, we *really* hope to have a further discussion with you to see if our responses solve your concerns. Thank you!!
>
> Sincerely,
>
> Authors

---

### Official Review · Reviewer_VvbL · 2021-11-01

**Correctness:** 3
**Technical Novelty And Significance:** 3
**Empirical Novelty And Significance:** 2
**Recommendation:** 5
**Confidence:** 3

**Main Review:**

Strength:

1) This paper brings new insights for the role of the “weights” in the network pruning.
2) This paper re-examines the established arguments with careful designed experiments over wide range of pruning ratios.
3) The proposed technique is clearly motivated by empirical observations and is very effective on MLP networks under large pruning ratios.
4) This paper is easy to follow.

Weakness:

1) Table 1,2 shows that fine-tuning a pruned network requires much more training time and tuning efforts to achieve a satisfying result. Although the author claims that OrthP can “complete” resolve the dynamical isometry issue, the pruned network converges still only after prolonged training epochs (90 and 900 epochs). Does that means the pruned networks weights are more difficult to optimize than random initialized ones, which contradicts the "the value of weights" claim?
2) The application area of the proposed OrthP is limited to MLP (Table 6). Although the author proposed a Strong L2 Regularization for more complicated networks, this section seems to be “unfinished” — the connection with dynamical isometry seems a little bit weak to me, can the authors provides more elaborations? This seems to be more of practical interests.
3) I have some confusions regarding the experiment designs and presented results (see the Questions section below).

Questions:

1)  For Table 1: In “Our rerun”,  are “Scratch” results trained under 120 epochs with a decayed LR schedule? For Table 2, what are the hyper-parameters for the “Scratch” results, are they carefully tuned?
2) Have the authors investigated the JSV of randomly initialized sparse networks? I am not sure if applying OrthP initialization or other orthogonal regularization methods on “Scratch” can also boost its performance. I believe this would be a fairer comparison to support "the value of weights" claim.
3) Table 5: Can the authors also provide the “Scratch” results and their hyper-parameters? I am asking because the reasons stated in Weakness 1) — if the “Scratch” result is comparable to OrthP without using prolonged epochs, then this means even when the dynamical isometry is recovered, the pruning selected weights need more training time to optimize than randomly initialized ones. Also, in Table 2, it looks like “Scratch” results work well under large PR, which is exactly the case in Table 5.
4) Related to 3), from Figure 2 and Table 5, it looks like OrthP takes the role of a large training epochs to recover dynamical isometry. It would be nice to see if OrthP works for non-prolonged epochs, e.g., < 90.

**Summary Of The Paper:**

This paper challenges an existing argument that “inheriting the weights of the pruned network is not necessary for fine-tuning”. This paper provides empirical evidence that the previous experiments are not carried out with proper learning rates and training epochs to ensure the network convergence.

The authors conjecture that fine-tuning pruned network requires larger learning rates and longer training epochs because its dynamical isometry is broken. As a result, large learning rates (if training for shorter epochs) or long training epochs are required for the recovery of dynamic isometry.

Then the authors propose OrthP to re-initialize the pruned weights to completely recover the dynamic isometry in simple MLP. With OrthP, the fine-tuning process is less sensitive to different learning rate and training epochs.


**Summary Of The Review:**

I like this paper in that it thoroughly investigates existing arguments on the value of pruning weights. The perspective of OrthP is not that novel, but is simple and effective on MLP under large pruning ratios.

My major concern is in the experiments design and results part. I would consider raising my score if my questions can be addressed.

---

> ### Author Response · Authors · 2021-11-19
> **Response to Review VvbL (R3) -- Part 3**
>
> The above results seem to put us in an awkward position since they are *against* what we claimed in the paper ("pruning has value"). But hold on, allow us to present another set of results -- We use **MLP-7-ReLU** (unpruned accuracy 98.16%) as evaluation network instead of **MLP-7-Linear** above. Clearly, here we want to see if the *non-linearity* can make any difference. Results are shown below:
>
> **Tab. R3-#3: Comparing L1 pruning to scratch training with two initialization schemes on MLP-7-ReLU**
>
> |  	|  PR 0.8 	| PR 0.9 	|
> |---	|---	|---	|
> | L1 pruning, 90 epochs, LR 0.01 | 96.75 (0.09)  |  94.76 (0.15) |
> | L1 pruning, *StrongReg*, 90 epochs, LR 0.01 | **96.98 (0.04)**  |  **95.10 (0.13)** |
> | Scratch (Kaiming_uniform) 	| 96.60 (0.16) 	| 94.64 (0.24)	|
> | Scratch (Orthogonal init) 	| 96.52 (0.17) 	| 92.56 (1.90) 	|
>
> where, for the L1 pruning, we replace the DI recovery method from OrthP to StrongReg since OrthP only works for *linear* MLP while here we are dealing with non-linear network.
>
> As seen, now, either using kaiming_uniform or orthogonal initialization, L1 pruning is *consistently better* than scratch training. The advantage is amplified when using StrongReg.
>
> A worthy question is: Why on the MLP-7-Linear, pruning does not beat orthogonal initialization but on MLP-7-ReLU, they beat? This is actually straightforward to see if we see pruning as *a* kind of initialization -- as mentioned above, on linear MLP  networks, orthogonal initialization is *proved to be the optimal*. That is, no other initialization can be better. Pruning is essentially also *a kind of initialization* for the subsequent fine-tuning, so it cannot beat the (optimal) orthogonal initialization, thus no value. But when it comes to the MLP-7-ReLU, orthogonal initialization is no longer optimal (as we mentioned, there is *no* method up to date that can achieve exact isometry for non-linear networks). Then, it is *likely* that pruning provides better initialization weights than orthogonal initialization or kaiming_uniform. It just turns out pruning really achieves this, and archives more with the help of StrongReg.
>
> There are at least 3 takeaways here:
> * First, The comparison above shows that whether pruning has value seriously hinges on the *network type, initialization scheme, and pruning ratios* (and maybe more). **Looking at pruning as a kind of initialization** as we do in this paper (and inspecting it with dynamical isometry) is actually a pretty good perspective, from which many results that appear inconsistent or random start to be explainable.
> * Second, as seen, pruning shows the value in the non-linear case but no value in the linear case. On the whole, we believe it is fair to say pruning *has* value considering the non-linear case is more practical.
> * Third, more profoundly, the above results actually suggest, **if dynamical isometry is fully recovered, pruning will (probably) have no value indeed because it cannot beat the initialization scheme that can achieve exact isometry.** Practically, up to date, for non-linear networks (not to mention BN, residuals, convolution), there has been no such method that can achieve exact isometry. Thus, it is still *likely* for pruning to be valuable at present -- In this regard, we recognize that our previous claim in the paper "pruning has value" is not rigorous enough, we update it to "**pruning has the potential to be valuable if the random initialization of scratch training cannot achieve exact isometry**".
>
> `R3-Q4`: Related to 3), from Figure 2 and Table 5, it looks like OrthP takes the role of a large training epochs to recover dynamical isometry. It would be nice to see if OrthP works for non-prolonged epochs, e.g., < 90.
>
> `R3-Response6`: This question is addressed in `R3-Response1` above, where we show the 30-epoch results using OrthP. It still works (with very marginal accuracy drop).
>
> Hope our (a little bit long) response can help clear your concerns. **If you have further questions regarding our feedback, just let us know!**

---

> ### Author Response · Authors · 2021-11-19
> **Response to Review VvbL (R3) -- Part 2**
>
> `R3-Q1`: For Table 1: In “Our rerun”, are “Scratch” results trained under 120 epochs with a decayed LR schedule? For Table 2, what are the hyper-parameters for the “Scratch” results, are they carefully tuned?
>
> `R3-Response3`: Yes, in Table 1 and 2, all the scratch models run by ourselves use the 120-epoch training strategy: SGD optimizer, weight decay 1e-4, LR schedule: 0:1e-1, 30:1e-2, 60:1e-3, 90: 1e-4, 105: 1e-5. This is adopted from the standard 90-epoch [PyTorch ImageNet example](https://github.com/pytorch/examples/blob/master/imagenet/main.py), just we add another 30-epoch training to *make sure the model is fully converged* (also mentioned in Appendix D). Since we use an even better strategy than the standard pytorch training strategy, we believe they *are* carefully tuned. But if you ask, are they tuned *really hard*? Then no, e.g., we definitely can use more tricks (like cosine LR schedule) to get even better results, but that would make the comparison non-standard and unfair to (Liu et al., 2019) -- note the training strategy used by the rethinking paper (Liu et al., 2019)) for ImageNet is also the 90-epoch [PyTorch ImageNet example](https://github.com/pytorch/examples/blob/master/imagenet/main.py), see [their github](https://github.com/Eric-mingjie/rethinking-network-pruning) ("For ImageNet, we use the official Pytorch ImageNet training code").
>
> `R3-Q2`: Have the authors investigated the JSV of randomly initialized sparse networks? I am not sure if applying OrthP initialization or other orthogonal regularization methods on “Scratch” can also boost its performance. I believe this would be a fairer comparison to support "the value of weights" claim.
>
> `R3-Response4`: Applying OrthP to randomly initialized networks is *the same* as orthogonal initialization proposed by (Saxe et al., 2014). Then this question actually reduces to "what about the training from scratch results using orthogonal initialization?". They are presented in our response `R3-Response5` below.
>
> `R3-Q3`: Table 5: Can the authors also provide the “Scratch” results and their hyper-parameters? I am asking because the reasons stated in Weakness 1) — if the “Scratch” result is comparable to OrthP without using prolonged epochs, then this means even when the dynamical isometry is recovered, the pruning selected weights need more training time to optimize than randomly initialized ones. Also, in Table 2, it looks like “Scratch” results work well under large PR, which is exactly the case in Table 5.
>
> `R3-Response5`: To answer this question, we first need to be very careful with what initialization scheme is used for the "scratch" training because we'll show they can make a *significant* difference. Specifically, we have two initialization schemes here: **(1) kaiming_uniform** (which is the [default initialization scheme in PyTorch](https://github.com/pytorch/pytorch/blob/68d8ab0cc60536db5a9af4c08ff39e43b252802f/torch/nn/modules/linear.py#L96) for conv and linear layers), **(2) orthogonal initialization**, proposed by (Saxe et al., 2014). Note that, orthogonal initialization is proved to be **optimal** in the sense of exact isometry *for linear MLP networks*.
>
> The hyper-parameters for training from scratch are: SGD optimizer, weight decay 1e-4, training for 90 epochs in total, initial LR 0.01, decayed by 0.1 at epoch 30, 60, batch size 100.
>
> Results are presented below:
>
> **Tab. R3-#2: Comparing L1 pruning to scratch training with two initialization schemes on MLP-7-Linear**
>
> |  	|  PR 0.8 	| PR 0.9 	|
> |---	|---	|---	|
> | L1 pruning, OrthP, 90 epochs, LR 0.01 	| 92.79 (0.03) 	| 92.77 (0.04) 	|
> | Scratch (Kaiming_uniform) 	| 92.60 (0.14) 	| 91.48 (0.23) 	|
> | Scratch (Orthogonal init) 	| 92.76 (0.03) 	| 92.76 (0.04) 	|
>
> As seen, L1 pruning beats the default PyTorch initialization (kaiming_uniform), while being on par with orthogonal initialization on PR (pruning ratio) 0.9. On PR 0.8, pruning does not show a statistically significant advantage. Again, the conclusion appears inconsistent here: if we compare L1 pruning with kaiming_uniform on PR 0.9, pruning has value; but if we compare with orthogonal initialization, pruning has no value. On PR 0.8, pruning cannot beat either of the two scratch results.  On the whole, fairly speaking, pruning has *no* value here since we find at least one kind of random initialization that can perform just as well.

---

> > ### Comment · Reviewer_VvbL · 2021-11-29
> > **Further Comments**
> >
> > Thanks for experiments in Tab R3-#2 and Tab R3-#3. I found them very interesting. Based on the authors' claim, the "value of pruning" mainly lies in nonlinear networks, where initializing with "pruning+strong reg" may achieve better isometry than existing initialization methods.
> >
> > As the authors later update their claim to be "pruning will (probably) have no values" in the linear case, the current paper arrangement becomes very confusing to me. I personally think it might be more appropriate to make "strong reg" the main contribution and take the linear case as a motivating observation.
> >
> > Also, several new claims in Section 5 need more rigorous supports (also pointed out by other reviewers). For example, 1) more evidence on "pruning+strong reg" can achieve better isometry than existing initialization and pruning methods, and 2) more analysis on why "pruning+strong reg" achieves better isometry.
> >
> > With these being said, I believe this paper has potential and is very interesting, but need further investigations on its claims. I will keep my original rating.

---

> > > ### Author Response · Authors · 2021-11-29
> > > **Thanks for Your Further Comments!**
> > >
> > > Dear Reviewer VvbL,
> > >
> > > *As the authors later update their claim to be "pruning will (probably) have no values" in the linear case, the current paper arrangement becomes very confusing to me.* -- Notably, we are updating our claim, making it more rigorous and specific, not *changing* it. Even without the MLP-7-Linear scratch results, in our original version, we still see examples against our "pruning has value" argument (e.g., on ImageNet, ResNet18, pruning ratio 0.9, 0.95, scratch is better than pruning). If our original paper arrangement is not "very confusing" (not mentioned previously), after adding the new section, it only makes our claim more specific and rigorous (not changing any of our major claims), thus should not be "very confusing" either.
> > >
> > > *I personally think it might be more appropriate to make "strong reg" the main contribution and take the linear case as a motivating observation* -- Although we still believe the major contribution of this paper is the analysis framework of dynamical isometry instead of StrongReg (note, only after we identify the dynamical isometry issue in filter pruning, then comes the necessity/legitimacy of dynamical isometry *recovery* by StrongReg. Currently, we are still at the "identifying problem" stage), we greatly thank you (and other reviewers) for the very constructive comments so far!
> > >
> > > Sincerely,
> > >
> > > Authors

---

> ### Author Response · Authors · 2021-11-19
> **Response to Review VvbL (R3) -- Part 1**
>
> Thank Reviewer VvbL (R3) for your very helpful comments! We address your concerns as follows. This response is a little bit long, so please be patient. Also, *if you have more concerns, we are more than happy to hear them! Just let us know!*
>
> `R3-Weakness1`: Table 1,2 shows that fine-tuning a pruned network requires much more training time and tuning efforts to achieve a satisfying result. Although the author claims that OrthP can “complete” resolve the dynamical isometry issue, the pruned network converges still only after prolonged training epochs (90 and 900 epochs). Does that means the pruned networks weights are more difficult to optimize than random initialized ones, which contradicts the "the value of weights" claim?
>
> `R3-Response1`: For the prolonged training epoch setting (90, 900 epochs), we intentionally use this setting (because we want to make sure the models are fully converged, as explained in the paper -- Page 7, "LR schedule setup"), which does *not* mean the model can *only* converge at this setting. With the normal number of training epochs on MNIST (30 epochs), it also converges -- we rerun the results of Tab. 5 with 30 epochs, results shown below. **All the results on MNIST in this rebuttal are averaged by 5 random runs (std shown in the parentheses)**.
>
> | OrthP+30 epochs 	|  PR 0.8 	| PR 0.9 	|
> |---	|---	|---	|
> | LR 0.01 	| 92.69 (0.05) 	| 92.66 (0.04) 	|
> | LR 0.001 	| 92.71 (0.07) 	| 92.72 (0.02) 	|
>
> As seen, they are pretty close (slightly worse) to the results of OrthP+90 and OrthP+900 epochs in Tab. 5 of the paper, but still *much better than not using OrthP*. Thus, our conclusion still holds when using this 30-epoch training strategy.
>
> We will still use the 90 epoch setting when reporting new results for this rebuttal.
>
> `R3-Weakness2`: The application area of the proposed OrthP is limited to MLP (Table 6). Although the author proposed a Strong L2 Regularization for more complicated networks, this section seems to be “unfinished” — the connection with dynamical isometry seems a little bit weak to me, can the authors provides more elaborations? This seems to be more of practical interests.
>
> `R3-Response2`: First, we present more elaborations as suggested -- StrongReg does not *explicitly* target isometry recovery, which may make R3 feel the connection is weak. But implicitly, it indeed recovers dynamical isometry -- Note dynamical isometry describes a nice state of the network that signals can propagate through it without serious magnitude explosion or attenuation. Since the weights in a network are *dependent* of each other, removing some of them will definitely hurt the isometry because it is based upon *all* the weights. When we use a strong regularization to push these unimportant parameters to zero, it explicitly makes the other parameters learn to *not rely on them*, that is, encouraging the gradients not to pass through these weights/neurons because they are going to be cleaned out. This is what we believe makes StrongReg effective in maintaining dynamical isometry. Its effectiveness is shown in Tab. 6 of the paper (also the results below). Besides, R1 has a similar question regarding why StrongReg is a DI recovery method. You may see our response above `R1-Response3` to see if it can help resolve your concern.
>
> StrongReg is very simple so we discuss it in just one page. This short length does not mean "unfinished".  We will add more explanation and results (e.g. those in `R1-Response3` above)  to the Appendix to explain more about StrongReg.
>
> Meanwhile, note that the main goal of this paper is *not* to propose a method to recover DI in pruning, but to explain how DI is playing a role in pruning and *make a formal response to the questioning of the value of pruning (Liu et al., 2019) (Crowley et al., 2018)*. Albeit being simple, StrongReg is proposed to be a good *starting point* that may inspire more advanced algorithms in the future.

---

> ### Author Response · Authors · 2021-11-27
> **Sincerely expecting further discussions with Reviewer VvbL (R3)**
>
> Dear Reviewer VvbL,
>
> We greatly thank you for the reviewing process so far! Given the ICLR final discussion deadline (11/29) is approaching, we *really* hope to have a further discussion with you to see if our responses solve your concerns. Thank you!!
>
> Sincerely,
>
> Authors

---

### Official Review · Reviewer_nWUS · 2021-11-03

**Correctness:** 2
**Technical Novelty And Significance:** 1
**Empirical Novelty And Significance:** 2
**Recommendation:** 3
**Confidence:** 4

**Main Review:**

# Strengths

1. The empirical observation that inheriting pretrained weights is better than training a sparse network from scratch if the fine-tuning phase is carefully tuned.
2. The empirical analysis on fine-tuning hyperparameters (LR, #epochs) might be useful to practitioners.

# Weaknesses

1. lack of novelty: connection to DI and pruning breaks DI is known [Lee et al 2020, analysed at initialization] and analysing for a pretrained network is straightforward and claiming it as a contribution is weak.

2. The hypothesis on page 7 connecting larger LR in fine-tuning and DI is dubious and not backed by any theory or experiments. In fig.2 plots mean JSV increases and that does not mean improved DI. As DI theory suggests to have mean JSV around 1.

3. The comparison against scratch does not seem fair. Usually, fine-tuning phase is much shorter compared to training from scratch but in this case fine-tuning is done for almost the same no of epochs as the scratch version. Is it possible that the scratch version could be tuned to improve the performance?

4. In section 4, the L2 regularization of pruned weights is regarded as a method for DI recovery which in my opinion is unsubstantiated.

**Summary Of The Paper:**

The paper argues pruning a pretrained network and finetuning is better than training a sparse network from scratch and tries to connect the fine-tuning phase properties to dynamical isometry (DI) literature. The empirical observation makes a case for inheriting the pretrained weights.

**Summary Of The Review:**

There are many unsubstantiated claims (or weak statements) in the paper.

---

> ### Author Response · Authors · 2021-11-20
> **Responses to Reviewer nWUS (R2)**
>
> Thank R2 for your very helpful comments! We address your concerns as follows.
>
> `R2-Weakness1`: lack of novelty: connection to DI and pruning breaks DI is known [Lee et al 2020, analysed at initialization] and analysing for a pretrained network is straightforward and claiming it as a contribution is weak.
>
> `R2-Response1`: Admittedly, we are using the *same tool* as  (Lee et al., 2020) -- dynamical isometry. But this does not mean "the contribution is weak". What matters more may not be what tool we use *but what we use it for*. In this regard, (Lee et al., 2020) used it for a *completely different* goal from ours: They used DI for proposing a better *pruning criterion* for pruning at initialization (trying to resolve that problem of over-pruning a layer in the global pruning ratio setting) while we use it to analyze how fine-tuning LR makes a difference in the pruning performance and further attempt to respond to the "no value of pruning" argument (Liu et al., 2019) (Crowley et al., 2018). Clearly, **we are towards completely different goals**. What you suggest is that the 1st difference ("Basic setting") we list in the paper is weak, yet we still have *3 more differences* (see the Related Work section on Page 3 "Pruning at initialization (PaI)"), which should sufficiently differentiate this paper from (Lee et al., 2020). Thus, we believe it is still fair for us to claim a contribution.
>
> `R2-Weakness2`: The hypothesis on page 7 connecting larger LR in fine-tuning and DI is dubious and not backed by any theory or experiments. In fig.2 plots mean JSV increases and that does not mean improved DI. As DI theory suggests to have mean JSV around 1.
>
> `R2-Response2`: Please see our response `How to properly look at the mean JSV metric for dynamical isometry` above, where we define how large of the mean JSV can be regarded as "around 1" for the MLP-7-Linear network. By the definition, the mean JSV range in Fig. 2 (2~5) can still be regarded as "around 1".
>
> `R2-Weakness3`: The comparison against scratch does not seem fair. Usually, fine-tuning phase is much shorter compared to training from scratch but in this case fine-tuning is done for almost the same no of epochs as the scratch version. Is it possible that the scratch version could be tuned to improve the performance?
>
> `R2-Response3`: "Usually, fine-tuning phase is much shorter compared to training from scratch" -- with all due respect, we think this statement may not be true either theoretically or practically:
> * Theoretically, there is no outstanding theory that demands the fine-tuning process *must* be short (`R2` may suggest specific references if you disagree). On the contrary, as a paper attempting to understand pruning more theoretically, this work actually suggests the *opposite* -- we should do a *long* finetuning because the broken dynamical isometry needs more training iterations to recover.
> * Practically, a short fine-tuning phase is not "usual" anymore. This was indeed the case for early pruning papers (such as the original L1-norm pruning paper (Li et al., 2017), which only used 20-epoch finetuning). But **more and more recent pruning methods actually prefer a *long* fine-tuning**: [*1]: 60 epochs, [*2]: 100 epochs, [*3]: 90 epochs, [*4]: 150 epochs, [*5]: 180 epochs.
> In fact, few of these papers explained why they need such a long fine-tuning. The finding in this paper can be a good principle to explain why they chose to do so.
>
> As for "*Is it possible that the scratch version could be tuned to improve the performance?*", Technically, it is always possible to get a better performance for the scratch version by adding more tricks (like cosine LR schedule). The point here is, we have to keep *fair* comparison. In this regard, the training strategy we adopt follows the standard [Pytorch ImageNet example](https://github.com/pytorch/examples/blob/master/imagenet/main.py), which is the *most standard* ImageNet training scheme. Also, this strategy was adopted by (Liu et al., 2019).
>
> `R2-Weakness4`: In section 4, the L2 regularization of pruned weights is regarded as a method for DI recovery which in my opinion is unsubstantiated.
>
> `R2-Response4`: Thanks for pointing out that it is not straightforward to understand StrongReg as a DI recovery method. Meanwhile, we also hope R2 can specifically point out *what makes you think so*. Besides, R1 also has a similar question. You may see our response `R1-Response3` above and see if it can lift your concern.
>
> --
> * [*1] Discrimination-aware channel pruning for deep neural networks, 2018, NeurIPS
> * [*2] Collaborative Channel Pruning for Deep Networks, 2019, ICML
> * [*3] Neural pruning via growing regularization, 2021, ICLR
> * [*4] Towards Compact CNNs via Collaborative Compression, 2021, CVPR
> * [*5] ResRep: Lossless CNN Pruning via Decoupling Remembering and Forgetting, 2021, ICCV
>
> Hope our response can help clear your concerns. **If you have further questions regarding our feedback, just let us know!**

---

> > ### Comment · Reviewer_nWUS · 2021-11-25
> > **Still some concerns remain**
> >
> > Thanks for the detailed response. Unfortunately, there are some concerns still remain.
> >
> > 1. From the empirical analysis one cannot "define" that the large of the mean JSV can be regarded as "around 1". This statement is stretching to connect to the existing DI theory. I would strongly suggest rephrasing that "in practice networks with JSV in the range (2-5) yield good empirical performance".
> >
> > 2. If a network from scratch can be tuned to match or exceed the performance of a pruned network, the 'value of pruning" becomes questionable. This also relates to the "Rectified Argument on Value of Pruning". It seems the main message of the paper has changed during the rebuttal phase due to the new experiments.
> >
> > 3. Related to the above, "pruning has the potential to be valuable if the random initialization of scratch training cannot achieve exact isometry" what is meant by "exact isometry" in this statement?
> >
> > 4. The L2 regularization does not directly incentives JSV to be close to 1, it simply encourages the magnitude of the weights to be small. From the experiments, if JSV becomes closer to 1 with strong L2 regularization, it is merely a side-effect. Further analysis is required to understand why is such behaviour emerging.

---

> > > ### Author Response · Authors · 2021-11-27
> > > **Responses to Left Concerns of Reviewer nWUS (R2)**
> > >
> > > Dear Reviewer nWUS,
> > >
> > > Thank you so much for the further comments! We have the following responses.
> > >
> > > 1. We will rephrase it as suggested. Meanwhile, although the empirical study is not equivalent to a proven theory (which we admitted), yet, (1) notably, existing DI theory studies the *trainability/initialization* of a network, where researchers have the freedom to decide what should be the best value of mean JSV (which is 1, obviously), however, in this work, we use mean JSV to study the behaviors of the network *during training*, not initialization. Thus we need to *adapt* it for our use. It is actually the best we have right now to analyze a practical network during training (if R2 has a better way to define "close to 1" for a practical neural network during training, you may specifically let us know), (2) the empirical study is informative and consistent with our expectation, thus pertinent and useful to our analyses.
> > >
> > > We are more than happy to make it more principled in the future, but considering this is still an early stage of a scientific exploration (our work is among the first one or two papers to examine network (filter) pruning via dynamical isometry), empirical studies are as important as theoretical proofs, too. Many theories start from empirical studies in the beginning. And note, the prior "pruning has no value" papers  (Liu et al., 2019) (Crowley et al., 2018) are also built upon empirical studies.
> > >
> > > 2. R2 stated "*It seems the main message of the paper has changed during the rebuttal phase due to the new experiments.*" -- The main message "pruning has value" does *not* change *on the whole* (see `Response to Review VvbL (R3) -- Part 3`: *On the whole, we believe it is fair to say pruning has value considering the non-linear case is more practical*) even when we have the MLP-7-Linear scratch results because the network is too simple to be practical. What we do in the "Rectified Argument on Value of Pruning" is to make our message more *rigorous* (not "changed"), because clearly, without specific context, discussing pruning has value or not is meaningless (the former is closer to practical cases, though). **The more important may not be to have a "one-sentence-fit-all" conclusion about the value of pruning (we cannot either, actually, as we've shown), but to know what is the reason under the hood answering for the mixed and seemingly inconsistent results**. This can greatly help us understand pruning in a more principled way. In this regard, (Liu et al., 2019) (Crowley et al., 2018) are purely empirical studies, while our work presents a faithful framework via dynamical isometry to explain the mixed behaviors of pruning. We believe this can be fairly regarded as a non-trivial contribution. R1 also agrees with this point: *Though some issues exists and DI may not be the ultimate solution to the question, `the paper provides valuable thoughts for this filed`*.
> > >
> > > 3. R2 stated: "*pruning has the potential to be valuable if the random initialization of scratch training cannot achieve exact isometry", what is meant by "exact isometry" in this statement?*" -- We have this statement because of the MLP-7-Linear scratch results, where OrthP can achieve exact isometry.  The "exact isometry" is defined the usual way: mean JSV equals 1.
> > >
> > > 4. R2 stated: "*From the experiments, if JSV becomes closer to 1 with strong L2 regularization, it is merely a side-effect. Further analysis is required to understand why is such behaviour emerging.*" -- Such behaviour emerging is because that network training itself can recover mean JSV (see page 7 of updated draft: "*In Tab. 4, one important fact is that, the mean JSV can recover itself without any extra help during finetuning, regardless of different setups*"). Just, in a normal manner of training, because of the weight dependency (as we explained in `R1-Response3`), the weight removal operation will damage the mean JSV anyway. Thus, we have to cut off this dependency. This is why we need StrongReg to peel off the weights that will be finally removed from the rest. Making them very small in magnitude is an effective way to cut off the dependancy. We believe such behavior is quite easy to understand with the above facts.
> > >
> > > Meanwhile, StrongReg serves as a starting point for inspiring more advanced DI recovery method. The major point of this paper is *not* DI recovery, but the proposed 4 hypotheses regarding the role of dynamcial isometry in filter pruning. This (plausible) theoretical-level understanding may be much more important than one specific DI recovery algorithm in our view. We hope R2 can pay more attention to our contribution there. Even if you think this paper may not be good enough for publication, if our responses so far help clear (some of) your concerns, could you kindly consider improving your score slightly?
> > >
> > > Sincerely,
> > >
> > > Authors

---

### Official Review · Reviewer_bhQQ · 2021-11-07

**Correctness:** 3
**Technical Novelty And Significance:** 3
**Empirical Novelty And Significance:** 3
**Recommendation:** 8
**Confidence:** 5

**Main Review:**

The paper is well written and organized. The effect of learning rate on fine-tuning stage of pruning is not thoroughly investigated and this paper provides an in-time and thorough study. My major concerns are as follows:

- Pruning as poor initialization: the concept of initialization needs to be cleared. Does the pruning in Fig 1 refer to weights pruning for both networks? It might be true if it is weights pruning as different pruning ratios correspond to the same weights dimensions with different values. However, filter pruning results in different architectures and different pruning results are not different initializations with respect to the same architecture.

- The dynamic isometry might explain the easiness of optimization, however, the relationship with the generalization is not quite clear. There might be other metrics such as flatness might show a similar trend as JSV. A comparison with other metrics for generalization could be more persuasive.

- The analysis with MLP-7 and MNIST might be too simple for practical usage, and this method does not generalize to more practical non-linear convolutional neural networks. The authors propose strong regularization helps, however, no explanation on why this method helps and how it connects with previous analysis. Does this method work for MLP-7 and how is it connected to DI?

**Summary Of The Paper:**

There are recent works questioning the value of inheriting weights in structured neural network pruning as it is empirically observed training from scratch can match or outperform finetuning a pruned model. This paper mainly includes three components: 1) the authors reinvestigated the problem and demonstrates that the conclusion is inaccurate because of improperly small finetuning learning rates. They show finetuning with pruned weights actually outperforms training from scratch, when larger learning rates and longer training epochs are adopted. 2)  the authors explored dynamical isometry (DI) to understand how finetuning LR affects the final performance. They show that weight pruning breaks dynamical isometry and finetuning can recover it and a larger LR can recover faster. 3) They proposed to fully recover dynamical isometry in fitler pruning before finetuning.


**Summary Of The Review:**

The paper carefully studies recent thoughts on the meaning of pruning and made some rigorous investigations. Though some issues exists and DI may not be the ultimate solution to the question, the paper provides valuable thoughts for this filed, including fair experimental comparison and new explanations. Therefore I recommend to accept.

---

> ### Author Response · Authors · 2021-11-21
> **Responses to Reviewer bhQQ (R1) -- Part 2**
>
> * **Intuition behind StrongReg**: Dynamical isometry describes a nice state of the network that signals can propagate through it without serious magnitude explosion or attenuation. Since the weights in a network are *dependent* on each other, removing some of them will definitely hurt the isometry because it is based upon *all* the weights. When we use a strong regularization to push these unimportant parameters to zero, it explicitly makes the other parameters learn to *not rely on them*, that is, encouraging the gradients not to pass through these weights/neurons because they are going to be cleaned out. This way, when the unimportant parameters are physically removed, it will incur much less damage to the left parameters, thus maintaining the network dynamical isometry well.
> * **Effectiveness on MLP-7-Linear**: We apply StrongReg to MLP-7-Linear at both PR 0.8 and PR 0.9. Results are presented below.
>
> |  	|  PR 0.8 	| PR 0.9 	|
> |---	|---	|---	|
> | L1 pruning, 90 epochs, LR 0.01	    | 91.36 (0.02)	| 87.81 (0.03) |
> | L1 pruning, 90 epochs, LR 0.001	    | 90.54 (0.02)	| 87.59 (0.01) |
> | L1 pruning, OrthP, 90 epochs, LR 0.01 | 92.79 (0.03)  | 92.77 (0.04) |
> | L1 pruning, OrthP, 90 epochs, LR 0.001 | 92.77 (0.03)  | 92.72 (0.03) |
> | L1 pruning, *Strong Reg*, 90 epochs, LR 0.01  | 92.80 (0.02) 	| 92.52 (0.04) 	|
> | L1 pruning, *Strong Reg*, 90 epochs, LR 0.001  | 92.80 (0.04)	| 92.48 (0.02) 	|
>
> As seen, similar to OrthP, StrongReg can rectify the test accuracy from the underrated ones to 92.48~92.80, close to the best possible performance (around 92.77). The left marginal gap at PR 0.9 may be because OrthP is the proven optimal isometry recovery method, while StrongReg is not mathematically optimal. But on the whole, StrongReg delivers very close performance to OrthP, demonstrating its effectiveness.
>
> To further show StrongReg really works by *maintaining dynamical isometry*, like Tab. #2, #3 above, we list the first 10-epoch mean JSV and test accuracies (at PR 0.9) and get the following summarized results:
>
> **Tab. R1-#2: Mean JSV of the first 10 epochs with different settings. Epoch 0 refers to the model just pruned, before any finetuning**
>
> | Epoch 	| 0 	| 1 	| 2 	| 3 	| 4 	| 5 	| 6 	| 7 	| 8 	| 9 	| 10 	|
> |---	|---	|---	|---	|---	|---	|---	|---	|---	|---	|---	|---	|
> | LR=0.01, w/ OrthP 	| 1.0000 	| 1.2318 	| 1.4144 	| 1.4277 	| 1.4017 	| 1.4709 	| 1.5171 	| 1.5551 	| 1.6082 	| 1.6538 	| 1.6648 	|
> | LR=0.001,  w/ OrthP 	| 1.0000 	| 1.5135 	| 1.6630 	| 1.7449 	| 1.8250 	| 1.8720 	| 1.9193 	| 1.9556 	| 1.9943 	| 2.0084 	| 2.0409 	|
> | LR=0.01, w/o OrthP 	| 0.0004 	| 0.6557 	| 0.8946 	| 1.0191 	| 0.9826 	| 1.0965 	| 1.1253 	| 1.2595 	| 1.3298 	| 1.2940 	| 1.4238  |
> | LR=0.001, w/o OrthP 	| 0.0004 	| 0.0004 	| 0.0006 	| 0.0014 	| 0.1103 	| 0.2765 	| 0.3501 	| 0.4320 	| 0.5167 	| 0.7478 	| 0.8501 	|
> | LR=0.01, *w/ StrongReg* 	| 3.0275 	| 2.1538 	| 2.0390 	| 1.0191 	| 1.9696 	| 2.1011 	| 1.9734 	| 2.0810 	| 2.0541 | 2.0563 | 2.0581 |
> | LR=0.001, *w/ StrongReg* 	| 3.0275	| 2.9691    | 2.9514    | 2.9921    | 2.9869    | 3.0101    | 3.0448    | 3.0516    | 3.0555 | 3.0441 | 3.0244 |
>
>
> **Tab. R1-#3: Test accuracy of the first 10 epochs with different settings. Epoch 0 refers to the model just pruned, before any finetuning**
>
> | Epoch 	| 0 	| 1 	| 2 	| 3 	| 4 	| 5 	| 6 	| 7 	| 8 	| 9 	| 10 	|
> |---	|---	|---	|---	|---	|---	|---	|---	|---	|---	|---	|---	|
> | LR=0.01, w/ OrthP 	| 9.74 	| 91.05 	| 91.39 	|91.33 	|91.37 	|91.74 	| 91.69 	| 90.74 	| 91.39 	|91.58 	|91.44 	|
> | LR=0.001,  w/ OrthP 	| 9.74 	|90.81 	|91.59 	|91.77 	|91.85 	|92.04 	|92.12 	|92.22 	|92.12 	|92.33 	|92.25 	|
> | LR=0.01, w/o OrthP 	| 9.74 	| 63.86 	| 79.96 	| 79.74 	| 80.06 	|85.79 	|85.82 	|86.11 	|86.45 	|86.53 	| 85.95 	|
> | LR=0.001, w/o OrthP 	| 9.74 	| 9.74 	| 9.74 	| 12.09 	|21.74 	|27.95 	|33.55 	|35.92 	|49.19 	|65.50 	|69.90 	|
> | LR=0.01, *w/ StrongReg* 	| 89.13 | 91.68 | 91.11 | 91.37 | 91.84 | 91.47 | 90.11 | 91.32 | 90.78 | 91.55 | 91.29 |
> | LR=0.001, *w/ StrongReg* 	| 89.13 | 92.03 | 91.84 | 92.13 | 92.05 | 91.85 | 92.00 | 92.02 | 92.09 | 92.08 | 92.08 |
>
> As seen, (1) First, at Epoch 0, the mean JSV is around 3 (which can be considered close to the exact isometry 1, by our analysis in `How to properly look at the mean JSV metric for dynamical isometry`) vs. 0.0004 when the dynamical isometry is not recovered. (2) Second, the test accuracy also has a high starting point: at Epoch 1, using StrongReg achieves 91.68 (LR 0.01) and 92.03 (LR 0.001) vs. 63.86 (LR 0.01) and 9.74 (LR 0.001) when the dynamical isometry is not recovered. In short, StrongReg plays a similar role to OrthP in maintaining dynamical isometry as well as test accuracy.
>
> Hope these new results can help you understand why StrongReg is a method to maintain dynamical isometry. **If you have further questions regarding our feedback, just let us know!**

---

> ### Author Response · Authors · 2021-11-21
> **Responses to Reviewer bhQQ (R1) -- Part 1**
>
> Thank R1 for your very helpful comments! We address your concerns as follows.
>
> `R1-Weakness1`: Pruning as poor initialization: the concept of initialization needs to be cleared. Does the pruning in Fig 1 refer to weights pruning for both networks? It might be true if it is weights pruning as different pruning ratios correspond to the same weights dimensions with different values. However, filter pruning results in different architectures and different pruning results are not different initializations with respect to the same architecture.
>
> `R1-Response1`: We only investigate filter pruning in this paper, so in Fig. 1, the pruning scheme refers to filter pruning. To prevent misunderstanding, we try to reiterate your concern here: *The weights after filter pruning at different pruning ratios are not different initializations with respect to the same architecture because filter pruning results in different architectures (specifically, layer widths)*. Filter pruning indeed leads to different widths, yet this is *not* contradicted with our "pruning as initialization" perspective: If we see filter pruning as zeroing out the full network instead of physically taking away the filters, then different pruning ratios correspond to different initialized weights (with different portions of zeros) *for the same architecture*. In this sense, pruning as initialization still holds even when we are dealing with filter pruning rather than weight element-wise pruning in this paper.
>
> `R1-Weakness2`: The dynamic isometry might explain the easiness of optimization, however, the relationship with the generalization is not quite clear. There might be other metrics such as flatness might show a similar trend as JSV. A comparison with other metrics for generalization could be more persuasive.
>
> `R1-Response2`: Great thanks for pointing out flatness as a potential metric for our investigation! Currently, in our mind, we only have the following way to utilize flatness: plot the loss contours of original unpruned model, pruned model (without finetuning), pruned model (after OrthP, without finetuning), pruned model with finetuning. Specifically, we are referring to the flatness visualization tool from the paper "Visualizing the Loss Landscape of Neural Nets" [2018, NeurIPS]. Yet, we do not find clear patterns from the plots. We'll keep exploring this direction in the future. Meanwhile, if the above way to utilize flatness is not what you actually meant, you may let us know what you think could be a better way to use the flatness metric.
>
> Besides, to further resolve your concern, note that, in today's neural networks, with abundant regularization techniques, generalization is generally well-aligned with optimization. One concrete example is, in Tab. #2 and #3 above (`[Response to All Reviewers] How to properly look at the mean JSV metric for dynamical isometry -- Part 2`), when mean JSV is low (implying a bad optimization), the *test* accuracy usually is low too (implying a bad generalization). Therefore, our current analyses in the paper should still be valid.
>
> `R1-Weakness3`: The analysis with MLP-7 and MNIST might be too simple for practical usage, and this method does not generalize to more practical non-linear convolutional neural networks. The authors propose strong regularization helps, however, no explanation on why this method helps and how it connects with previous analysis. Does this method work for MLP-7 and how is it connected to DI?
>
> `R1-Response3`: Thank R1 for letting us know it is not straightforward to see the connection between StrongReg and DI. Here we first explain its intuition behind and further show its effectiveness on MLP-7-Linear:

---

### Author Response · Authors · 2021-11-20
**[Response to All Reviewers] How to properly look at the mean JSV metric for dynamical isometry (1/2)**

We use these abbreviations to mention different reviewers: `R1`: bhQQ, `R2`: nWUS, `R3`: VvbL, `R4`: YYie.

**Thanks to all the reviewers for your constructive comments!** We find several reviewers are curious about how the mean JSV is related to dynamical isometry. Thanks for letting us know this critical concept is not clearly treated in our paper. Here we elaborate more. All the reviewers are strongly encouraged to look at this section.

DI (dynamical isometry) is defined by mean JSV close to 1 in (Saxe et al., 2014) (we are aware that, rigorously in (Saxe et al., 2014), DI describes the distribution of *all* JSVs. Mean JSV is only an average sketch of the distribution. But this approximation is accurate enough for analysis, also used by (Lee et al., 2020) (in fact, we are directly inspired by (Lee et al., 2020) for using this metric). In other words, **if a network has mean JSV close to 1, we can say this network has dynamical isometry**.

Then a non-trivial technical question is: **When we deal with practical DNNs in the real world, how close is the so-called “close to 1”**? To our best knowledge, there is no outstanding theory to *quantify* this (if any reviewer disagrees, you may suggest specific references), so we resort to empirical analysis, specifically on the MLP-7-Linear network used in our paper.

 In Tab. #1 below, we present the mean JSV of MLP-7-Linear network on MNIST under different pruning ratios, along with their accuracies before and after fine-tuning.

**Tab. #1. Mean JSV and accuracies of MLP-7-Linear on MNIST under different pruning ratios. Each result is randomly run for 3 times. We report the mean accuracy and (std).**

| pruning ratio (PR) 	| 0 	| 0.1 	| 0.2 	| 0.3 	| 0.4 	| 0.5 	| 0.6 	| 0.7 	| 0.8 	| 0.9 	|
|---	|---	|---	|---	|---	|---	|---	|---	|---	|---	|---	|
| mean JSV 	| 2.4987 	| 1.7132 	| 0.9993 	| 0.5325 	| 0.2711 	| 0.1180 	| 0.0452 	| 0.0151 	| 0.0040 	| 0.0004 	|
| Accuracy before finetuning (%) 	| 92.77 	| 91.35 	| 78.88 	| 62.21 	| 32.14 	| 11.47 	| 9.74 	| 9.74 	| 9.74 	| 9.74 	|
| Accuracy after finetuning (%) 	| / 	| 92.82 (0.05) 	| 92.80 (0.04) 	| 92.80 (0.01) 	| 92.77 (0.01) 	| 92.77 (0.02) 	| 92.77 (0.00) 	| 92.78 (0.02) 	| 91.37 (0.03) 	| 87.82 (0.03) 	|

As seen, there is a clear trend: larger pruning ratio, smaller mean JSV, lower accuracy (before or after finetuning). Particularly note *the mean JSV range where the pruned network can be finetuned back to the original accuracy (92.77%)*, which is **>=0.0151**. This means, for networks with mean JSV >= 0.0151, in spite that their immediate accuracies (without finetuning) can be distinct (e.g., 91.35% at PR 0.1 vs. 9.74% at PR 0.7), intrinsically, they are equivalently potential after finetuning. By this “*trainable with equivalent potential*” rule, **if a mean JSV lies in the range of >=0.0151, we can regard it as “close to 1” because they can do just as well as 1** (we do not need to pay as much attention to what is the upper limit of mean JSV in defining "close to 1", because in practice, a normally trained network *rarely* present a *very large* mean JSV by our observation, but is *quite likely* to have a *very small* mean JSV (0.0004 at PR 0.9 in Tab. #1 is an example).

Then, for Fig. 2 in our paper, the most important point we want to convey is actually in the *early stage (epoch < 10)*. We have realized that Fig. 2 is not well-plotted in a proper scale, so here we use a table instead, which should be more straightforward to see:

---

> ### Author Response · Authors · 2021-11-20
> **[Response to All Reviewers] How to properly look at the mean JSV metric for dynamical isometry (2/2)**
>
>
>  **Tab. #2. Mean JSV of the first 10 epochs with different settings. Epoch 0 refers to the model just pruned, before any finetuning**
>
> | Epoch 	| 0 	| 1 	| 2 	| 3 	| 4 	| 5 	| 6 	| 7 	| 8 	| 9 	| 10 	|
> |---	|---	|---	|---	|---	|---	|---	|---	|---	|---	|---	|---	|
> | LR=0.01, w/ OrthP 	| 1.0000 	| 1.2318 	| 1.4144 	| 1.4277 	| 1.4017 	| 1.4709 	| 1.5171 	| 1.5551 	| 1.6082 	| 1.6538 	| 1.6648 	|
> | LR=0.001,  w/ OrthP 	| 1.0000 	| 1.5135 	| 1.6630 	| 1.7449 	| 1.8250 	| 1.8720 	| 1.9193 	| 1.9556 	| 1.9943 	| 2.0084 	| 2.0409 	|
> | LR=0.01, w/o OrthP 	| 0.0004 	| 0.6557 	| 0.8946 	| 1.0191 	| 0.9826 	| 1.0965 	| 1.1253 	| 1.2595 	| 1.3298 	| 1.2940 	| 1.4238 	|
> | LR=0.001, w/o OrthP 	| 0.0004 	| 0.0004 	| 0.0006 	| 0.0014 	| 0.1103 	| 0.2765 	| 0.3501 	| 0.4320 	| 0.5167 	| 0.7478 	| 0.8501 	|
>
> **Tab. #3. Test accuracy of the first 10 epochs with different settings. Epoch 0 refers to the model just pruned, before any finetuning**
>
> | Epoch 	| 0 	| 1 	| 2 	| 3 	| 4 	| 5 	| 6 	| 7 	| 8 	| 9 	| 10 	|
> |---	|---	|---	|---	|---	|---	|---	|---	|---	|---	|---	|---	|
> | LR=0.01, w/ OrthP 	| 9.74 	| 91.05 	| 91.39 	|91.33 	|91.37 	|91.74 	| 91.69 	| 90.74 	| 91.39 	|91.58 	|91.44 	|
> | LR=0.001,  w/ OrthP 	| 9.74 	|90.81 	|91.59 	|91.77 	|91.85 	|92.04 	|92.12 	|92.22 	|92.12 	|92.33 	|92.25 	|
> | LR=0.01, w/o OrthP 	| 9.74 	| 63.86 	| 79.96 	| 79.74 	| 80.06 	|85.79 	|85.82 	|86.11 	|86.45 	|86.53 	| 85.95 	|
> | LR=0.001, w/o OrthP 	| 9.74 	| 9.74 	| 9.74 	| 12.09 	|21.74 	|27.95 	|33.55 	|35.92 	|49.19 	|65.50 	|69.90 	|
>
> As seen in Tab. #2, without OrthP, the mean JSV of LR 0.001 arises very slowly compared to LR 0.01. With OrthP, the mean JSV of LR 0.001 is not much different from LR 0.01 (note their mean JSV starting point is already 1 because OrthP can recover the pruned model to exact isometry). *Particularly note how the mean JSV trend in Tab. #2 correlates with test accuracy trend in Tab. #3*. This apparent correlation is another reason we adopt mean JSV for investigation.
>
> ---
> The mean JSV scale can vary case by case (up to the specific network, dataset, training optimizer, etc.) and there is no outstanding theory to quantify it. Therefore, looking at it properly really *needs some rule-of-thumb here*, e.g., for mean JSV = 2 vs. 0.001, we are pretty sure the former is probably better; but for mean JSV = 5 vs. 2, we cannot really have any reliable conclusion about which is better (in this sense, `R4` asked "*can increasing mean JSVs from ~1 to ~5 (as in Fig 2) be deemed meaningful?*", our answer is also *no*. Yet, in the early phase of Fig. 2, mean JSV arises from 0.0004 to around 1. This can be fairly deemed meaningful).
>
> Hope our empirical study above can help reviewers understand how the mean JSV can be an (unperfect but still useful) proxy for dynamical isometry and especially how to look at the mean JSV improvement at a proper scale. Since this is an empirical study, not a proven theory, if any reviewer has further questions concerning it, we are more than happy to take them and see if we can have a more faithful metric to measure dynamic isometry and further improve our work.

---

### Author Response · Authors · 2021-11-23
**Thanks to All Reviewers! Draft Updated.**

Dear Reviewers,

We have updated the draft according to your comments. The changed parts are highlighted in **orange color** in the new draft pdf. There are the following main changes. All these changes are also in the response boxes below but more formally. All reviewers are *strongly* recommended to take a look.

* Add a new section "*How to Properly Look at the Mean JSV Metric for Dynamical Isometry*" in Appendix D, since properly using mean JSV is a key to understanding this paper.
* Remove the poorly-plotted Fig. 2 of mean JSV and test accuracy. Instead, use tables (Tabs. 4 and 5 on page 7) to present the first 10-epoch mean JSV and and test accuracy, where the correlation between mean JSV and test accuracy is more obvious. The analysis text and the boxed hypothesis are also updated accordingly, on page 7. (Fig. 2 is still kept in the Appendix, page 15, for reference)
* Add a new section "*Rectified Argument on Value of Pruning*" (page 9) to address the newly-brought concern in Tab. 7, where the results show pruning has *no* value. We analyze the reason and update our claims on the value of pruning to a more rigorous version: "`pruning has the potential to be valuable if the random initialization of scratch training cannot achieve exact isometry`”. Three key takeaways are also presented.
* Add a new section in Appendix E to clarify why the proposed StrongReg is related to dynamical isometry -- it works just like OrthP: (1) improves the mean JSV at the early stage of fine-tuning, (2) improves the test accuracy, (3) closes up the accuracy difference between LR 0.01 and 0.001.

Hope our efforts so far can help resolve your concerns! *We are more than happy to keep in communication with you in the following rebuttal period! Let us know if you have any further questions*. Thanks!

Sincerely,

Authors

---

### Decision · Program_Chairs · 2022-01-20

**Decision:**

Reject

**Comment:**

The paper's primary contributions are:
* Contrary to previous claims, the authors empirically show that inheriting the weights after pruning can be beneficial when using *larger* fine-tuning learning rates than previously done.
* As an explanation, the authors provide suggestive results showing that pruning breaks dynamical isometry, which they claim explains why larger learning rates are needed.
* They propose a regularization-based technique to recover dynamical isometry on modern residual CNNs.

Generally, reviewers were positive about the ideas in the paper, however, even after the rebuttal 3/4 reviewers did not find the arguments were clear or strongly supported yet. One issue that came up several times is a request for more investigation of StrongReg+pruning. At this time, I have to recommend rejection, but I encourage the authors to follow up on the reviewers suggestions and submit to a future venue.